# Integrated Transcriptomic and Metabolomics Analyses Reveal Molecular Responses to Cold Stress in Coconut (*Cocos nucifera* L.) Seedlings

**DOI:** 10.3390/ijms241914563

**Published:** 2023-09-26

**Authors:** Lilan Lu, Weibo Yang, Zhiguo Dong, Longxiang Tang, Yingying Liu, Shuyun Xie, Yaodong Yang

**Affiliations:** 1Hainan Key Laboratory of Tropical Oil Crops Biology/Coconut Research Institute, Chinese Academy of Tropical Agricultural Sciences, Wenchang 571339, China; lulilan1234@163.com (L.L.); yangweibo623@126.com (W.Y.); dongzg@catas.cn (Z.D.); tanglx@catas.cn (L.T.); 2School of Earth Sciences, China University of Geosciences, Wuhan 430074, China; yingying1795@126.com

**Keywords:** coconut, transcriptomic, metabolomics, cold stress

## Abstract

Coconut is an important tropical and subtropical fruit and oil crop severely affected by cold temperature, limiting its distribution and application. Thus, studying its low-temperature reaction mechanism is required to expand its cultivation range. We used growth morphology and physiological analyses to characterize the response of coconuts to 10, 20, and 30 d of low temperatures, combined with transcriptome and metabolome analysis. Low-temperature treatment significantly reduced the plant height and dry weight of coconut seedlings. The contents of soil and plant analyzer development (SPAD), soluble sugar (SS), soluble protein (SP), proline (Pro), and malondialdehyde (MDA) in leaves were significantly increased, along with the activities of superoxide dismutase (SOD), peroxidase (POD), and catalase (CAT), and the endogenous hormones abscisic acid (ABA), auxin (IAA), zeatin (ZR), and gibberellin (GA) contents. A large number of differentially expressed genes (DEGs) (9968) were detected under low-temperature conditions. Most DEGs were involved in mitogen-activated protein kinase (MAPK) signaling pathway-plant, plant hormone signal transduction, plant–pathogen interaction, biosynthesis of amino acids, amino sugar and nucleotide sugar metabolism, carbon metabolism, starch and sucrose metabolism, purine metabolism, and phenylpropanoid biosynthesis pathways. Transcription factors (TFs), including WRKY, AP2/ERF, HSF, bZIP, MYB, and bHLH families, were induced to significantly differentially express under cold stress. In addition, most genes associated with major cold-tolerance pathways, such as the ICE-CBF-COR, MAPK signaling, and endogenous hormones and their signaling pathways, were significantly up-regulated. Under low temperatures, a total of 205 differentially accumulated metabolites (DAMs) were enriched; 206 DAMs were in positive-ion mode and 97 in negative-ion mode, mainly including phenylpropanoids and polyketides, lipids and lipid-like molecules, benzenoids, organoheterocyclic compounds, organic oxygen compounds, organic acids and derivatives, nucleosides, nucleotides, and analogues. Comprehensive metabolome and transcriptome analysis revealed that the related genes and metabolites were mainly enriched in amino acid, flavonoid, carbohydrate, lipid, and nucleotide metabolism pathways under cold stress. Together, the results of this study provide important insights into the response of coconuts to cold stress, which will reveal the underlying molecular mechanisms and help in coconut screening and breeding.

## 1. Introduction

Coconut (*Cocos nucifera* L.) is a tropical fruit and oil cash crop that is essential on a global scale and is rich in nutrients, including fats, proteins, sugars, fatty acids, and amino acids [1,2]. Coconuts taste sweet. The flesh has the effects of tonifying deficiency, strengthening, clearing summer heat, and quenching thirst [3]. Coconut, in the palm family, is mainly planted in tropical and subtropical coastal areas [4,5]. The most suitable temperatures for growth are between 27 and 32 °C, with an average annual temperature of 29 °C. Even if the average annual temperature for one month is 18 °C, the yield decreases significantly. Average temperatures lower than 15 °C cause the fall of flowers and fruits, cracking of fruits, and yellowing of leaves [6,7]). Thus, low temperatures substantially affect the growth and yield of coconut. The demand for coconut has been increasing, prompting research into the mechanism of its cold resistance and the cultivation of a new cold-resistant variety [8].

Low temperature is a major environmental stress factor that substantially affects plant growth, development, quality, and geographical distribution [9,10,11] and can change morphological, biochemical, and physiological characteristics [10]. Cold stress impedes enzyme activity, photosynthesis, and other biochemical processes, leading to the accumulation of reactive oxygen species (ROS), which leads to oxidative damage and membrane instability [12]. Oxidative stress induced by cold stress is the product of the accumulation of ROS. ROS are produced for signaling purposes and are anoxic byproducts of aerobic metabolism. For specific responses to gene expression, ROS trigger signal transduction pathways, such as mitogen-activated protein kinase (MAPK) signal transduction pathways. Overly toxic ROS are cleared by enzymes, such as superoxide dismutase (SOD), catalase (CAT), and peroxidase (POD) [13,14]. During cold stress, the leakage of the plasma membrane destroys this enzyme system, leading to metabolic disorders. This further leads to a decrease in plant energy supply, inhibits photosynthesis and biomass production, and causes plant death [12,15,16].

To cope with cold stress, plants adopt various physiological, biochemical, and molecular regulations, including changes in membrane lipid composition, induction of enzyme activity, changes in gene expression, hormone control, signal transduction, and osmotic regulation mechanisms to maintain plant cell homeostasis [17]. Plant adaptation to cold stress varies by species [16]. Moreover, plants in tropical and subtropical regions are generally sensitive to cold stress [18,19].

Plant responses to cold stress involve complex regulatory networks that bring about a wide range of physiological and metabolic changes, such as the activation of SOD and the accumulation of soluble sugars and low-molecular-weight substances [20]. This further aids in ROS removal and relieves osmotic pressure to maintain cell homeostasis [21]. Under cold stress, a series of molecular changes occur, along with changes in metabolic processes [22]. When plants are subjected to low-temperature stress, several cold response genes participate in signal transduction, and regulatory pathways are induced, resulting in the accumulation of specific osmoregulatory substances [9]. Under low-temperature stress, a significant increase in the accumulation of protective substances, such as soluble sugar and protein, has been observed in many plants [23,24]. Studies have shown that soluble sugar levels in different plants increase significantly under low-temperature stress, such as in citrus (*Citrus reticulata*) [25], red spruce [23], and *Pinus halepensis* [26]. The concentration of proline is a critical factor in cold tolerance. As an osmoregulatory compound, proline affects wheat [27] and influences ROS homeostasis, plasma membrane integrity, and cold resistance of the crops. In addition, as a global signal transduction regulator, the plant hormone abscisic acid (ABA) stabilizes membrane structure, regulates stomatal movement, and controls osmotic stress tolerance in plants through transcriptional regulation of downstream stress-related genes [28]. The application of exogenous ABA enhanced the cold tolerance of *Magnolia liliiflora* [29] and tomatoes [30]. There is also increasing evidence that jasmonic acid (JA) is involved in the regulation of plant cold tolerance [31]. These reports show the complexity of plant responses to low temperatures. However, metabolic changes in response to cold stress vary by plant species [32].

Altered gene expression in response to low temperatures is a commonly used strategy for managing plant chilling injuries [33]. Many components involved in cold response signaling pathways have been isolated and characterized, including messenger molecules (e.g., Ca^2+^), Ca^2+^-associated protein kinases, and several key transcription factors (TFs) [34]. One of the best-studied examples is the plant ICE-C-repeat/DREB binding factors (CBF)-COR signaling module [35]. C-cyclic peptide binding factor/dehydration reaction element binding protein (CBFs/DREB), a member of the AP2/ethylene reaction factor (AP2/ERF) family, plays a central role in cold acclimatization [36]. Transcription levels of CBF are substantially up-regulated by inducible factor CBF expression protein (ICE), a MYC-type alkaline helix-loop-helix family TF; CBF then activates downstream COR gene expression by binding to cis-elements in the promoter [37]. In addition, many cold-resistant proteins and protective substances (e.g., soluble sugar and proline) are synthesized in plant cells, which regulate osmotic potential and maintain membrane integrity [38,39].

Transcriptome analysis is a highly effective method to determine cold response genes in many crops, such as *Brassica napus* [40] and *Brassica juncea* [41]. With the rapid development of high-throughput sequencing and mass spectrometry technologies, advances in transcriptomics and metabolomics have enabled global molecular and physicochemical analysis of the regulatory mechanisms of plant cold tolerance [22,42]. Transcriptome changes using RNA sequencing (NA-SEQ) have revealed plant responses to cold stress in *Arabidopsis* [43], winter rapeseed [20], rapeseed [40], peanut [44], and *Argyranthemum frutescens* [45]; WRKY, NAC, MYB, AP2/ERF, and bZIP have been reported as the most abundant TF families of low-temperature stress in many species [40,46]. In addition, various metabolites, including ABA, carbohydrates, amino acids, and polyamines, have been used to investigate cold-stress responses in several species [44,47,48,49]. Transcriptomic and metabolomic analysis and mining of these big data facilitate the exploration of the mechanism of plant cold resistance [9,50]. Comparative transcriptomic analysis of two peanut cultivars (NH5 and FH18) with contrasting cold resistance identified several cold response TFs, including bHLH, MYB, and NAC [51]. In addition, lipidomics analysis of these two varieties indicated that membrane lipid and fatty acid metabolism contributed indirectly to the cold resistance of peanut [52].

A comprehensive analysis of transcriptomics and metabolomics is essential for understanding the complex regulatory networks involved in plant cold-stress responses. Under cold stress, the combined transcriptomics and metabolomics of *Xanthoceras sorbifolia* showed enhanced metabolism of amino acids and sugars [32]. Low-temperature stress induced significant changes in the transcriptome and metabolome. Key pathways related to ABA/JA signaling and proline biosynthesis play an important role in regulating cold resistance in wheat [9]. Most differentially expressed genes (DEGs) and differentially accumulated metabolites (DAMs) are mainly enriched in different carbohydrates and amino acid metabolisms. Among them, starch, sucrose, and phenylalanine metabolism were significantly enriched, which played a crucial role in the adaptation of *Brassica napus* to cold stress [53]. Most DEGs are involved in amino acid biosynthesis, plant hormone signaling, and MAPK signaling pathways. In addition, metabolomics analysis showed that the contents of free polyamines (PAs), plant hormones, and osmotic solution mainly included increased putrescine, spermidine, spermidine, ABA, JA, raffinose, and proline when pepper was responding to cold stress. Importantly, regulation of the ICE-CBF-COR pathway through Ca^2+^, MAPK, and ROS signaling plays a crucial role in regulating the response of pepper to cold stress [54]. Moreover, comprehensive analysis of the transcriptome and metabolome showed that most of the genes and metabolites involved in carbohydrate metabolism, the TCA cycle, and flavonoid biosynthesis were up-regulated in cold-tolerant pepper cultivars under cold stress [55]. This comprehensive approach has also been widely used to study the cold-stress response of peanut [44], pumpkin [56], and tobacco [10,57] crops.

Despite the substantial progress in understanding cold-stress responses in food crops and ornamentals, knowledge of this response in palm woody oil crops is limited. In plant biology, it is crucial to have a comprehensive understanding of the cold-stress responses of various plant species. Studies have examined cold-stress responses of the woody oil crop of the palm family, oil palm [58,59,60], and the cold resistance of coconut [61,62]. However, the molecular mechanisms underlying the signaling pathways and related gene networks require further research owing to the complexity of cold resistance traits in crops.

In this study, we aimed to explore the transcriptomic and metabolomic responses of coconut seedlings under cold stress and identify the relationship between gene expression and corresponding metabolite accumulation and depletion. Moreover, we aimed to evaluate the main growth and physiological indexes of coconut seedlings, explore the transcriptome and metabolome changes in coconut under low-temperature stress, identify the signaling pathways and regulatory networks related to cold tolerance, and discuss the response mechanism of the coconut under cold stress. Identifying the underlying molecular mechanisms of the coconut hypothermia response will facilitate the development of future coconut hardy varieties.

## 2. Results

### 2.1. Physiological Differences in Response to Cold Stress

The effects of cold stress on the growth of coconut seedlings were evaluated using pot experiments, in which seedlings were exposed to cold stress (5 °C) (low-temperature treatment [LT]) and room temperature (25 °C) (the control [CK]). The LT directly affected the growth of coconut seedlings (Figure 1a) and significantly reduced their plant height and dry weight (*p* < 0.05): after 30 d of the LT, the plant height and dry weight of coconut seedlings were 31.08% and 49.07% less, respectively, than those of the CK (Appendix A). In addition, under cold stress, the soil and plant analyzer development (SPAD) value was significantly lower (55.14%) than that of the CK (*p* < 0.01) (Figure 1b). SS and SP contents increased significantly under cold stress, peaked on day 20, and gradually decreased from day 20 to 30, generally increasing by 41.33% and 41.38%, respectively (*p* < 0.05) (Figure 1c,d). Thus, cold stress seriously affected the growth, photosynthetic index, and quality of coconut seedlings and had significant effects on the physiological characteristics of the coconut seedlings (Figure 2). Under cold stress, malondialdehyde (MDA) and PRO contents increased significantly; they showed trends similar to those of prolonged cold stress, with an increase of 34.64% and 31.42%, respectively (*p* < 0.05) (Figure 2a,b). The activities of CAT, POD, and SOD in the leaves of coconut seedlings under cold stress were significantly higher (*p* < 0.05), 25.35%, 48.73%, and 33.08%, respectively, than those of CK, and under the prolonged cold stress, they displayed similar trends. However, the increase in SOD activity was lower than that in POD and CAT (Figure 2c–e). In addition, cold stress had significant effects on endogenous hormone levels in the leaves of coconut seedlings. Under processes of cold stress, ABA content increased slowly within 10 d of the LT, but rapidly from day 20 to 30. During the LT, ABA content increased significantly by 51.95% (*p* < 0.01) (Figure 2f), and the contents of IAA, ZR, and GA increased within 20 d of the LT, peaked on 20 d, and gradually decreased from days 20 to 30. The contents of IAA, ZR, and GA after LT were thus significantly higher, by 27.43%, 24.57%, and 22.02%, respectively (*p* < 0.05) (Figure 2g–i), than those of CK. Thus, cold stress seriously affected the physiological function indexes and hormone levels of coconut seedlings.

### 2.2. Transcriptome Analysis

#### 2.2.1. Evaluation of Transcriptome Sequencing Data

After sequencing quality control, 38.42 GB of clean data were obtained from six samples (i.e., three biological duplicate samples from two experimental treatments), with the percentage of Q30 bases in each sample no less than 93.48%. According to the statistical results, the efficiency of comparison between the reads of each sample and the reference genome ranged between 94.04% and 94.87% (Appendix A). Fragments per kilobase of transcript per million fragments mapped (FPKM) > 1 was used as the threshold to determine gene expression, and the FPKM value in the CK_30_ samples was higher than that in the LT_30_ samples (Appendix A). The correlation analysis of FPKM values between the LT_30_ and CK_30_ samples is shown in Appendix A. Principal component analysis (PCA) showed that the LT_30_ and CK_30_ samples were clustered separately, indicating significant differences in gene expression between sample groups. The three bioreplicated samples under the LT_30_ and CK_30_ conditions were strictly clustered together, indicating the high biorepeatability of the samples treated in each group (Appendix A). Appendix A shows the volcano maps with significantly up-regulated and down-regulated differences between the two groups.

RNA-seq detected 23,795 genes with appropriate FPKM values in the comparison of LT_30_ versus CK_30_ (Appendix A). Under cold stress, there were 11,591 DEGs in the leaves of coconut seedlings (|log_2_FC| ≥ 1 with a false discovery rate [FDR] < 0.01): 5669 up-regulated and 5922 down-regulated (Appendix A, Appendix A). Moreover, 9968 DEGs were functionally annotated: 4805 up-regulated and 5163 down-regulated (Appendix A, Appendix A). A total of 8524 DEGs (Figure 3a and Appendix A) were annotated to the Gene Ontology (GO) items, and there were 55 significantly enriched GO items (FDR [error detection rate] limited the criterion to ≤0.05; DEGs No. ≥ 4): 21 biological process (BP) categories, 18 cellular component (CC) categories, and 16 molecular function (MF) categories (Figure 3a). The most abundant gene in the BP category belonged to the “cellular process (4289 DEGs)”, followed by the “metabolic process (4105 DEGs)” and the “single-organism process (2628 DEGs)”. In the CC category, the highest number of genes were found in “cell (4602 genes)” and “cell part (4602 genes)”, followed by “organelle (3478 genes)” and “membrane (2559 genes)”. The main terms of the MF category were “binding (4370 genes)” and “catalytic activity (3780 genes)” (Figure 3a and Appendix A). In the comparison of the LT_30_ and CK_30_, in the top 20 GO enrichment items, the BP category included “RNA modification (291 DEGs)”, “macromolecule metabolic process (230 DEGs)”, “macromolecule modification (172 DEGs)”, “cellular aromatic compound metabolic process (164 DEGs)”, “cellular nitrogen compound metabolic process (163 DEGs)”, and “organic cyclic compound metabolic process (163 DEGs)”. Most of the DEGS in these GO-enriched items were significantly down-regulated (Figure 3b and Appendix A, Appendix A). In the CC category, the GO significantly enriched items comprised “intracellular membrane-bounded organelle (638 DEGs)” and “mitochondrion (197 DEGs)”. Most of their DEG expression levels were down-regulated (Figure 3c and Appendix A, Appendix A).“Zinc ion binding (364 DEGs)”, “binding (282 DEGs)”, “catalytic activity (276 DEGs)”, “hydrolase activity (224 DEGs)”, “organic cyclic compound binding (181 DEGs)”, “heterocyclic compound binding (179 DEGs)”, and “structural constituent of ribosome (168 DEGs)” were significantly enriched in MF, with the majority of the DEGs significantly down-regulated (Figure 3d and Appendix A, Appendix A).

For Kyoto Encyclopedia of Genes and Genomes (KEGG) analysis, a total of 3452 DEGs were assigned to 132 KEGG pathways involved in metabolism, genetic information processing, organismal systems, environmental information processing, and cellular processes (Appendix A, Appendix A). Moreover, 1665 up-regulated DEGs were assigned to 127 KEGG pathways and 1787 down-regulated DEGs to 128 KEGG pathways, where the most top 20 important pathways by all DEGs mainly were “MAPK signaling pathway-plant (191 DEGs)”, “biosynthesis of amino acids (169 DEGs)”, “amino sugar and nucleotide sugar metabolism (99 DEGs)”, “glycerophospholipid metabolism (82 DEGs)”, “circadian rhythm-plant (61 DEGs)”, “phenylalanine, tyrosine and tryptophan biosynthesis (45 DEGs)”, “steroid biosynthesis (31 DEGs)”, and “lipoic acid metabolism (6 DEGs)” (*p* < 0.05) (Figure 4a and Appendix A, Appendix A). In addition, most DEGs were also enriched in pathways containing “plant-pathogen interaction (318 DEGs)”, “plant hormone signal transduction (265 DEGs)”, “carbon metabolism (165 DEGs)”, “starch and sucrose metabolism (120 DEGs)”, “glycolysis/gluconeogenesis (85 DEGs)”, “purine metabolism (78 DEGs)”, and “phenylpropanoid biosynthesis (66 DEGs)” (Appendix A). Under cold stress, KEGG enrichment analysis identified six significantly enriched pathways by up-regulated DEGs, including “plant-pathogen interaction (186DEGs)”, “carbon metabolism (105 DEGs)”, and “circadian rhythm-plant (44 DEGs)” (*p* < 0.05) (Figure 4b and Appendix A, Appendix A). Moreover, most of the up-regulated DEGs were also associated with the pathways of “plant hormone signal transduction (151DEGs)”, “MAPK signaling pathway-plant (98 DEGs)”, “biosynthesis of amino acids (78 DEGs)”, “starch and sucrose metabolism (54 DEGs)”, “glycolysis/gluconeogenesis (54 DEGs)”, “purine metabolism (41 DEGs)”, and “glycerophospholipid metabolism (41 DEGs) (Appendix A), whereas seven pathways contained “amino sugar and nucleotide sugar metabolism (62 DEGs)”, “phenylalanine, tyrosine and tryptophan biosynthesis (33 DEGs)”, “steroid biosynthesis (25 DEGs)”, and “lipoic acid metabolism (6 DEGs)” by down-regulated DEGs (*p* < 0.05) (Figure 4c and Appendix A, Appendix A). Moreover, most of the down-regulated DEGs were also associated with the pathways of “plant-pathogen interaction (132 DEGs)”, “plant hormone signal transduction (114 DEGs)”, “MAPK signaling pathway-plant (93 DEGs)”, “biosynthesis of amino acids (91 DEGs)”, “starch and sucrose metabolism (66 DEGs)”, “carbon metabolism (60 DEGs)”, “phenylpropanoid biosynthesis (45 DEGs)”, “glycerophospholipid metabolism (82 DEGs)”, “purine metabolism (37 DEGs)”, and “glycolysis/gluconeogenesis (31 DEGs)”. The above results suggest that these pathways played a key role in the coconut response to cold stress (Appendix A).

#### 2.2.2. The Major Regulator Genes of Significantly Enriched KEGG Pathways in Response to Cold Stress in Coconut

We further analyzed the major regulator genes of significantly enriched KEGG Pathways, including “plant hormone signal transduction”, “MAPK signaling pathway-plant”, “plant-pathogen interaction”, “biosynthesis of amino acids”, “amino sugar and nucleotide sugar metabolism”, “glycerophospholipid metabolism”, “circadian rhythm-plant”, “carbon metabolism”, “starch and sucrose metabolism”, “glycolysis/gluconeogenesis”, “purine metabolism”, “phenylpropanoid biosynthesis”, “steroid biosynthesis”, and “lipoic acid metabolism” in LT_30_ vs. CK_30_ (Figure 5, Appendix A). The information of these DEGs and log_2_FC have been summarized in Appendix A. A total of 265 DEGs were associated with plant hormone signal transduction, in which 57% (151 DEGs) were up-regulated; the genes encoding RR23, EIN3FP1, SAUR32, IPN2, ETR2, BHLH41, SAPK3, and LECRK2 were significantly up-regulated; while the expression of 43% of DEGs (114 DEGs) was down-regulated. The expressions of BHLH25, At4g37250, Os01g0656200, XTH22, PYL10, ABI5, LRK10, Os08g0500300, MYC4, and SHR1 genes were significantly down-regulated (Figure 5, Appendix A). In the MAPK signaling pathway-plant pathway, 191 DEGs were detected, of which 51% (98 DEGs) were up-regulated, and the genes encoding CML46, EIN3FP1, WRKY24, WRKY51, WRKY65, ETR2, BHLH41, WRKY14, SAPK3, MPK5, CP1, and LECRK2 were significantly up-regulated. However, 49% of DEGs (93 DEGs) were down-regulated, and the genes encoding PERK1, BHLH25, RBOHF, Os01g0656200, BHLH57, PYL10, LRK10, Os08g0500300, MYC4, At1g05700, At1g67720, and VQ20 were significantly down-regulated (Figure 5, Appendix A). In total, 99 DEGs were related to amino sugar and nucleotide sugar metabolism, among which 37 DEGs were up-regulated, along with the genes encoding CHIT3, CHIA, GAE1, GEPI48, BDX, HXK4, FRK1, and LOC109505297. However, 62 DEGs were down-regulated, and the expressions of genes encoding GAUT9, HXK3, BXL6, GAE6, BHLH35, UAM1, Os03g0268400, FLN2, PMI2, PMI1, CHIT3, and UEL-1 were significantly down-regulated (Figure 5, Appendix A). A total of 85 DEGs involved in the Glycolysis/Gluconeogenesis pathway were detected, 54 of which were up-regulated. The expression of genes encoding PK, AKR4C10, FBA5, AKR4C9, TPIP1, HXK4, GAPC, PGM1, and PFK3 were significantly up-regulated. As against, 31 DEGs were down-regulated, similar to the genes encoding HXK3, GAPCP2, gpmA1, PGPE, EMB3003, LTA2, PCK1, and TCTP (Figure 5, Appendix A). Overall, 78 DEGs were detected in purine metabolism, of which 41 were up-regulated, and the genes encoding PK, XDH, RSH1, RSH2, At2g26230, ASE1, andLOC105046338 were significantly up-regulated. On the contrary, 37 DEGs were down-regulated, and the genes encoding RE2, RPTNT1-94, RNR1, APK31, APK3, NMAT4, and DDB_G0275467 were significantly down-regulated (Figure 5, Appendix A).

#### 2.2.3. The Core Genes and Transcription Factors in Response to Cold Stress in Coconut

ICE-CBF-COR is the most important reaction pathway for plants at low temperature. The genes involved in cold−regulated protein, cold-responsive protein kinase, and temperature-induced lipocalin−1 were up-regulated under cold stress (Figure 6). One TF ICE1(SCRM) gene encoding COCN_GLEAN_10023355 was up-regulated; however, another TF ICE1(SCRM) gene encoding COCN_GLEAN_10008268 was down-regulated. Six DREB/CBF transcription factors in the field of dehydration-responsive element-binding protein (DREB1E (COCN_GLEAN_10009450), expressions of COCN_GLEAN_10008816 and COCN_GLEAN_10001620), DREB1C, DREB2C, and DREB1G were significantly up-regulated. However, the expressions of two DREB/CBF transcription factors (DREB3 and DREB1F) were significantly down-regulated. In the late embryogenesis abundant proteins, D-34 and LEA5 were significantly up-regulated, but At5g17165 and LEA14-A were down-regulated (Figure 6). During the cold treatment, the core DEGs in cold response, including auxin-responsive proteins and factors (45), calcium-binding proteins (26), calmodulin-related proteins (24), calcium-dependent protein kinase (13), calcineurin B-like protein (5), calcineurin B-like protein and CBL-interacting protein kinase (21), mitogen-activated protein kinase (MAPK) cascades (31), and LIM domain-containing proteins (3), were identified (Figure 6, Appendix A). The DEGs information and log_2_FC have been summarized in Appendix A. In auxin-responsive proteins and the factors, 24 DEGs were up-regulated, and the genes involved in the auxin-responsive proteins (SAUR32, ARGOS, SAUR71, Os09g0247700, IAA10, SAUR36) and the auxin response factors (ARF1, ARF8, ARF12, ARF6, ARF9, ARF18) were significantly up-regulated (*p* < 0.05). However, 21 DEGs were down-regulated, in which the auxin-responsive proteins (IAA10, IAA16, IAA6, SAUR71, PIN1C, PIN3A, AUX28, LAX2, SAUR50, SAUR36) and the auxin response factors (ARF19, ARF17, ARF2) were significantly down-regulated. In calcium-binding proteins, 85% of DEGs (22 DGEs) were up-regulated, mainly including CML46, CML38, CML19, CML41, PBP1, At1g02270, CP1, CML30, CML36, KIC, CML29, CML35, CML15, CML14, and CML11, whereas CML7, CML25, CML21, and CBP were significantly down-regulated. In calmodulin-related proteins, 18 DEGs were up-regulated, including CML2, CAMBP25 (COCN_GLEAN_10018164, COCN_GLEAN_10000260), CBP60F, CML3, CBP60E, CAM1, CBP60A, CML30, CAMTA3, and CAMRLK in significant up-regulation, while six DEGs, CAMTA4, NADKC, and CAMBP25 (COCN_GLEAN_10000461) were significantly down-regulated. In calcium-dependent protein kinase, 70% of DEGs (nine DEGs), such as CPK20, CPK3, CPK10, CPK4, and CPK1, were significantly down-regulated. In calcineurin B-like protein, two DEGs encoding CBL9 (CUFF17.251.1) and CBL7 were up-regulated, but three encoding CBL1 (CUFF34.779.2, CUFF5.369.1) and CBL9 (COCN_GLEAN_10019851) were down-regulated. In CBL-interacting protein kinase, 75% of DEGs (12 DEGs) were up-regulated, and CIPK18, CIPK5, CIPK6, CIPK14, CIPK12, CIPK19, and CIPK22 were significantly up-regulated, while 25% of DEGs encoding CIPK2, CIPK23, and CIPK1 were down-regulated. In mitogen-activated protein kinase (MAPK), five DEGs encoding MPK5, MPK14, MPK3, MPK1 (COCN_GLEAN_10018606 and COCN_GLEAN_10018607), and CBL7 were up-regulated, with MPK5 and MPK14 being significantly up-regulated; however, four DEGs encoding MPK9 (COCN_GLEAN_10007573, COCN_GLEAN_10003541, COCN_GLEAN_10003954) and MPK10 were down-regulated. In mitogen-activated protein kinase kinase (MAPKK, or MKK), MKK1 and MKK2 were up-regulated, and MKK1 was significantly up-regulated, but six DEGs encoding MKK5, MKK9, MKK3, and MKK4 (COCN_GLEAN_10020255, COCN_GLEAN_10022613, COCN_GLEAN_10005503) were down-regulated, and MKK5 was significantly down-regulated. In mitogen-activated protein kinase kinase kinase (MAPKKK, or MEKK), 12 DEGs were up-regulated, among which, 8 DEGs encoding MAPKKK18 (COCN_GLEAN_10015847, COCN_GLEAN_10018889), MAPKKK3 (COCN_GLEAN_10000563, CUFF3.335.1) and MEKK1(Cocos_nucifera_newGene_19341, COCN_GLEAN_10013851, COCN_GLEAN_10003127, COCN_GLEAN_10024549) were significantly up-regulated. However, only two DEGs encoding MEKK(NPK1) and MEKK(YDA) were down-regulated. In LIM domain-containing proteins, WLIM2B was up-regulated; however, two genes encoding WLIM1 (Cocos_nucifera_newGene_4820 and Cocos_nucifera_newGene_4822) were down-regulated (Figure 6, Appendix A). TFs plays an important role in regulating the expression of stress response genes under cold stress. In the transcriptomic analysis, several types of TFs were found, which mainly belonged to WRKY (43), AP2/ERF (42), HSF (18), GATA (16), bZIP (15), Trihelix (14), MYB (13), bHLH (3), NFY (4), NAC (2), CHR (2), MADS(2), RL9 (2), and other TFs (6) (Figure 6, Appendix A). Under low-temperature treatment, 67% of WRKY family members (29) were up-regulated, and WRKY53, WRKY51, WRKY24, WRKY65, WRKY41, WRKY6, and WRKY14 were significantly up-regulated. While 33% of WRKY TFs (14) were down-regulated, WRKY28, WRKY49, and WRKY22 were significantly down-regulated. A total of 74% of AP2/ERF family members (31) were up-regulated, among which ERF017, ERN1, ERF4, ERF018, ERF109, ERF071, ERF008, At4g13040, ERF053, ERF5, ERF1B, and ERF3 were significantly up-regulated; however, 26% of AP2/ERF TFs (11) were down-regulated, with ERF034, ERF061, ERF026, and ERF019 in significantly down-regulated expressions. In total, 89% of HSF family members (16) were up-regulated, and HSFB1, HSFA3, HSFC1A, HSFA4B, HSFB2B, HSFC2B, and HSFA5 were significantly up-regulated. Overall, 81% of GATA family members (13) were down-regulated, and GATA28, GATA2, GATA22, GATA9, and GATA4 were significantly up-regulated; however, GATA5, GATA8, and GATA26 were down-regulated. In total, 80% of bZIP family members (12) were up-regulated, and TGA10, RISBZ2, BZIP44, BZIP53, and BZIP5 were significantly up-regulated; however, BZIP1-A (COCN_GLEAN_10015700, CUFF1.1372.1) and BZIP68 were significantly down-regulated. A total of 50% of Trihelix family members (7) were up-regulated, and GT-3B was significantly up-regulated; however, 50% of Trihelix TFs (7) were down-regulated, of which ASIL1, ASIL2, and ASR3 were significantly down-regulated. A total of 38% of MYB family members (5) were up-regulated, and IPN2 (COCN_GLEAN_10006593, COCN_GLEAN_10006962), MOF1 (COCN_GLEAN_10006112), PHL11, and PHL7 were significantly up-regulated; however, 62% of MYB TFs (8) were down-regulated, with MOF1 (COCN_GLEAN_10014415), EFM, PHL8, MPH1, and PHL5 in significantly down-regulated expressions. In the bHLH family, two DEGs encoding BHLH41 (COCN_GLEAN_10013682, COCN_GLEAN_10011130) were significantly up-regulated; however, a DEGs encoding BHLH41 (COCN_GLEAN_10013779) was significantly down-regulated. In the NFY family, NFYC4 was significantly up-regulated, but NFYC6, NFYB9, and NFYA7 were significantly down-regulated. In the CHR family, CHR28 was up-regulated, but CHR27 was down-regulated. In the NAC family, NAC056 was significantly up-regulated, but NAC032 was down-regulated. In addition, the expression levels of TFs such as MADS22, MADS15, RL9, POSF21, and YY1 were significantly down-regulated under low-temperature conditions (Figure 6, Appendix A).

#### 2.2.4. qRT-PCR Validation

Nineteen DEGs involved in the MAPK signaling pathway-plant, amino sugar and nucleotide sugar metabolism, plant hormone signal transduction, carbon metabolism, starch and sucrose metabolism, and biosynthesis of amino acids were selected to verify the accuracy of the RNA-seq data using qRT-PCR. Under low temperatures, the expressions of the ETR2, MPK5, and WRKY24 genes associated with the MAPK signaling pathway-plant were significantly up-regulated, and BHLH25 was significantly down-regulated. The GAE1 and UXS2 genes related to amino sugar and nucleotide sugar metabolism were significantly up-regulated, and BHLH35 was significantly down-regulated. The XTH22, DPBF3, and SCL9 genes related to plant hormone signal transduction were significantly up-regulated, and IAA10 was significantly down-regulated. Expressions of the CYSC and SHM7 genes related to carbon metabolism were significantly up-regulated. The GLU3, GLC1, and SUS4 genes related to starch and sucrose metabolism were significantly up-regulated. The expressions of the FBA5 and CYSC genes related to the biosynthesis of amino acids were significantly up-regulated, and those of HAPS-1 were significantly down-regulated. These results suggest that a good correlation with FPKM values was obtained from the RNA-seq data (R^2^ = 0.987), which confirms the reliability of gene expression data measured by RNA-seq (Appendix A).

### 2.3. Metabolome Analysis

#### 2.3.1. Metabolite Profiles of Coconut in Response to Cold Stress

Metabolomics analysis was performed to characterize metabolites of coconut in response to cold stress. A total of 423 metabolites in positive-ion mode and 225 metabolites in negative-ion mode were identified in the comparison of cold treatment and control samples (Appendix A). DAMs were defined by the criteria of |log_2_FC| ≥ 1, *p* < 0.05, and VIP > 1. In the positive-ion model, a total of 206 DAMs were identified, while in the negative-ion mode, a total of 97 DAMs. Heat maps of DAMs between the two groups also revealed responses to cold stress at the metabolite level and successfully divided the test samples into two broad categories (Figure 7a,e). According to principal component analysis (PCA), PC1, PC2, and PC3 accounted for 64.69%, 13.32%, and 10.64% of metabolites in the positive-ion mode, respectively (Figure 7b), and for 64.40%, 15.66%, and 9.16% of metabolites in the negative-ion mode, respectively (Figure 7f), indicating a significant separation between the LT_30_ and CK_30_ groups. In the OPLS-DA analysis, the separation of the two groups was considerably good (Appendix A). This indicates good repeatability between the three experimental (LT_30_) or three control (CK_30_) samples and a significant degree of separation between LT and CK samples. In the positive-ion mode, a total of 95 (up-regulated)/111 (down-regulated) DAMs (Appendix A, Appendix A) were identified, including phenylpropanoids and polyketides (16%); lipids and lipid-like molecules (19%); benzenoids (9%); organoheterocyclic compounds (14%); organic oxygen compounds (8%); organic acids and derivatives (12%); nucleosides, nucleotides, and analogues (4%); organic nitrogen compounds (2%); alkaloids and derivatives (1%); and others (15%) (Figure 7d, Appendix A). A total of 38 (up-regulated)/59 (down-regulated) DAMs were identified in the negative-ion mode (Appendix A, Appendix A), including phenylpropanoids and polyketides (25%); lipids and lipid-like molecules (8%); benzenoids (9%); organoheterocyclic compounds (12%); organic oxygen compounds (11%); organic acids and derivatives (3%); nucleosides, nucleotides, and analogues (6%); and others (14%) (Figure 7h, Appendix A).

#### 2.3.2. Differentially Accumulated Metabolites (DAMs) in Response to Low Temperatures

In the positive-ion mode, the top 10 up-regulated DAMs were l-Lysine (log_2_FC = 15.93), apiin (log_2_FC = 9.30), delta-Tocotrienol (log_2_FC = 9.17), mesoridazine (log_2_FC = 8.05), *N*-Acetyl-d-lactosamine2,4-Diaminobutyric acid (log_2_FC = 7.77), 7-Keto-8-aminopelargonic acid (log2FC = 7.60), bis(2-ethylhexyl) phthalate (log2FC = 7.26), harpagoside (log2FC = 7.25), cytarabine (log_2_FC = 7.17), and 2-Ethyl-4-hydroxy-5-methyl-3(2*H*)-furanone (log_2_FC = 6.93). In contrast, the top 10 down-regulated DAMs were hydrocortisone (log_2_FC = −16.64), 17-beta-Estradiol-3-glucuronide (log_2_FC = −14.29), 6-Ketoprostaglandin E1 (log_2_FC = −13.94), 4-Hydroxycinnamic acid (log_2_FC = −9.77), 2,4-Diaminobutyric acid (log_2_FC = −9.44), lamivudine (log_2_FC = −8.38), 5-Hydroxymethylcytidine (log_2_FC = −8.30), scytalone (log_2_FC = −7.60), picolinic acid (log_2_FC = −7.19), and PE (16:0/0:0) (log_2_FC = −7.17) (Appendix A). In the negative-ion mode, the top 10 up-regulated DAMs were d-FRUCTOSE (log_2_FC = 19.10), gemcitabine (log_2_FC = 18.55), cromolyn (log_2_FC = 16.68), phenylpyruvate (log_2_FC = 16.43), shikimate (log_2_FC = 6.29), biochanin A (log_2_FC = 5.61), propynoic acid (log_2_FC = 5.35), narcissin (log_2_FC = 5.15), threoninyl-Proline (log_2_FC = 5.09), and teniposide (log_2_FC = 5.07). On the contrary, the top 10 down-regulated DAMs were *N*1-Methyl-2-pyridone-5-carboxamide (log_2_FC = −21.28), triflupromazine (log_2_FC = −18.01), 5,2′-*O*-dimethyluridine (log_2_FC = −17.82), serinyl-Proline (log_2_FC = −16.01), N-Glycolylneuraminic acid (log_2_FC = −15.69), baccatin III (log_2_FC = −13.85), quercitrin (log_2_FC = −11.98), l-Aspartyl-l-phenylalanine (log_2_FC = −9.31), 1,3-Dicaffeoylquinic acid (log_2_FC = −7.92), and sucrose (log_2_FC = −7.54) (Appendix A).

To explore the physiological processes of these DAMs, KEGG annotation and analysis were performed. In the positive-ion mode (Figure 7c and Appendix A, Appendix A), the 52 DAMs were mainly enriched in “Amino acid metabolism” (including “Lysine biosynthesis”, “Tryptophan metabolism”, “Cysteine and methionine metabolism”, “Lysine degradation”, “Histidine metabolism”), “Metabolism of other amino acids”(“d-Amino acid metabolism”, “Cyanoamino acid metabolism”), “Biosynthesis of other secondary metabolites” (“Isoquinoline alkaloid biosynthesis”, “Biosynthesis of various plant secondary metabolites”, “Flavone and flavonol biosynthesis”, “Flavonoid biosynthesis”, “Phenylpropanoid biosynthesis”, “Biosynthesis of various alkaloids”), “Lipid metabolism” (Glycerophospholipid metabolism”, “Arachidonic acid metabolism”), “Membrane transport” (“ABC transporters”), “Metabolism of cofactors and vitamins”(“Nicotinate and nicotinamide metabolism”, “Biotin metabolism”, “Glutathione metabolism), and “Translation”(“Aminoacyl-tRNA biosynthesis”) pathways. In the negative-ion mode (Figure 7c and Appendix A, Appendix A), the 34 DAMs were mainly enriched in “Amino acid metabolism” (containing “Phenylalanine, tyrosine, and tryptophan biosynthesis”), “Biosynthesis of other secondary metabolites” (“Biosynthesis of various plant secondary metabolites”, “Flavone and flavonol biosynthesis”, “Flavonoid biosynthesis”, “Phenylpropanoid biosynthesis”, “Isoflavonoid biosynthesis”), “Carbohydrate metabolism” (“Galactose metabolism”, “Starch and sucrose metabolism”), “Membrane transport” (“ABC transporters”), “Metabolism of cofactors and vitamins” (“Folate biosynthesis”), and “Nucleotide metabolism” (Pyrimidine metabolism) pathways. Notably, several metabolic pathways, such as biosynthesis of other secondary metabolites and membrane transport, were also in the negative-ion mode and included “Biosynthesis of various plant secondary metabolites”, “Flavone and flavonol biosynthesis”, “Flavonoid biosynthesis”, “Phenylpropanoid biosynthesis”, “Isoflavonoid biosynthesis”, and “ABC transporters”. The above results of KEGG enrichment pathway analysis of DAMs suggest that these metabolic pathways played important roles in stress signaling and response in coconut.

### 2.4. Correlation Analysis between DEGs and DAMs

Based on the 11,591 DEGs and 303 DAMs (206 in positive-ion mode, 97 in negative-ion mode), screening was conducted according to the correlation coefficient (CC) and correlation *p*-value, with the screening thresholds being |CC| > 0.70 and *p* < 0.05, respectively. Among the 9 quadrants, the patterns of DEGs and DAMs were consistent in quadrants 1, 3, 7, and 9 (Appendix A), and the regulation of genes and metabolites was positively or negatively correlated (*p* < 0.01, R^2^ > 0.8). The correlation coefficient matrix heat maps of DAMs and DEGs also showed a similar trend (Appendix A). Combined transcriptome and metabolome analysis in the positive-ion mode revealed 47 common KEGG enrichment pathways from 3452 DEGs to 56 DAMs (Figure 8a, Appendix A). In the negative-ion model, 31 common enrichment KEGG pathways were identified based on 3452 DEGs and 34 DAMs (Figure 8b, Appendix A); the significant enrichment KEGG pathways of DEGs and DAMs of positive and negative ions were as follows: “amino acid metabolism”, “metabolism of other amino acids”, “biosynthesis of other secondary metabolites”, “metabolism of cofactors and vitamins”, “carbohydrate metabolism”, “lipid metabolism”, “nucleotide metabolism”, “translation”, and “membrane transport” pathways, which mainly included “lysine biosynthesis (ko00300)”, “tryptophan metabolism (ko00380)”, “cysteine and methionine metabolism (ko00270)”, “lysine degradation (ko00310)”, “histidine metabolism (ko00340)”, “phenylalanine, tyrosine and tryptophan biosynthesis (ko00400)”, “tyrosine metabolism (ko00350)”, “2-Oxocarboxylic acid metabolism (ko01210)”, and “biosynthesis of amino acids (ko01230)” in the “amino acid metabolism” pathway; “cyanoamino acid metabolism (ko00460)” and “glutathione metabolism (ko00480)” in the “metabolism of other amino acids” pathway; “isoquinoline alkaloid biosynthesis (ko00950)”, “flavone and flavonol biosynthesis (ko00944)”, “flavonoid biosynthesis (ko00941)”, “phenylpropanoid biosynthesis (ko00940)”, “tropane, piperidine and pyridine alkaloid biosynthesis (ko00960)”, “stilbenoid, diarylheptanoid and gingerol biosynthesis (ko00945)”, and “isoflavonoid biosynthesis (ko00943)” in the “biosynthesis of other secondary metabolites” pathway; “vitamin B6 metabolism (ko00750)”, “ubiquinone and other terpenoid-quinone biosynthesis (ko00130)”, “nicotinate and nicotinamide metabolism (ko00760)”, “biotin metabolism (ko00780)”, and “folate biosynthesis (ko00790)” in the “metabolism of cofactors and vitamins” pathway; “galactose metabolism (ko00052)”, “starch and sucrose metabolism (ko00500)”, and “carbon metabolism (ko01200) ”in the “carbohydrate metabolism” pathway; glycerophospholipid metabolism (ko00564)” and “arachidonic acid metabolism (ko00590)” in the “lipid metabolism” pathway; “pyrimidine metabolism (ko00240)” in the “nucleotide metabolism” pathway; “aminoacyl-tRNA biosynthesis (ko00970)” in the “translation” pathway; and “ABC transporters (ko02010)” in the “membrane transport” pathway (Figure 8, Appendix A). However, some pathways that played important roles in the regulation of cold stress were found significantly enriched only by DEGs (*p* < 0.05), including “biosynthesis of amino acids (ko01230)”, “amino sugar and nucleotide sugar metabolism (ko00520)”, “MAPK signaling pathway-plant (ko04016)”, “plant hormone signal transduction (ko04075)”, “steroid biosynthesis (ko00100)”, “lipoic acid metabolism (ko00785)”, “phenylalanine, tyrosine and tryptophan biosynthesis (ko00400)”, and “Alanine, aspartate and glutamate metabolism (ko00250)” (Figure 8).

#### 2.4.1. Amino Acid Metabolism Considerably Contributes to Cold Stress in Coconut

In plants, some amino acids, such as proline and arginine, as osmotic substances, are critical regulatory elements in stress responses. Interestingly, KEGG pathways involved in amino acid metabolism were found to be abundant in the expression levels of both metabolites and genes in LT_30_ versus CK_30_ under cold stress. Based on the correlation analysis between the two groups (*p* < 0.05) and the enrichment KEGG pathway of genes and metabolites, we constructed the correlation network of DEGs and DAMs that may be involved in cold stress (Figure 9 and Appendix A, Appendix A). In the lysine biosynthesis pathway, the genes involved in amino acid metabolism encoding AGD2, LOC105058024, COCN_GLEAN_10015782, DHDPS2, AK1, DAPF, and DHDPS2 were down-regulated under cold stress, and were significantly negatively correlated with the up-regulated metabolites, such as l-Lysine (+) and l-Saccharopine (+), but were significantly positively correlated with the down-regulated l-2-Aminoadipate adenylate (+). However, the up-regulated expression of the LYSA gene was significantly positively correlated with l-Lysine (+) and l-Saccharopine (+) and significantly negatively correlated with l-2-Aminoadipate adenylate (+). In lysine degradation, 29 DEGs were detected that were involved in amino acid metabolism, and 15 DEGs were up-regulated, also showing significant positive correlation with l-Lysine (+) and l-Saccharopine (+). Among them, the genes encoding FRK1, ASHR3, LOC103697661, CCACVL1_23055, PAP4, ASHR3, and LKR/SDH were significantly up-regulated, while 14 DEGs were down-regulated and were significantly negatively correlated with l-Lysine (+) and l-Saccharopine (+), in which the genes encoding TCTP and ALDH3F1 were significantly down-regulated. In tyrosine metabolism, a total of 24 DEGs were found involved in amino acid metabolism, among which 7 DEGs were up-regulated and showed a significant positive correlation with 2,5-Dihydroxybenzaldehyde (+), but were significantly negatively correlated with 4-Hydroxycinnamic acid (+), in which primary amine oxidase (PAOD), At1g62810, prudu_014267, and At5g42250 were significantly up-regulated; however, 17 DEGs were down-regulated and showed a significant negative correlation with 2,5-Dihydroxybenzaldehyde (+), but were significantly positively correlated with 4-Hydroxycinnamic acid (+), and the genes encoding AATG/AAT, LOC105044482, TYDC, and PPOD were significantly down-regulated. In the tryptophan metabolism pathway, a total of 25 DEGs involved in amino acid metabolism were detected, of which 9 DEGs were up-regulated and showed a significant positive correlation with cinnavalininate (+), but a significant negative correlation with 5-Hydroxy-l-tryptophan (+), picolinic acid (+), and indoleacetaldehyde (−), and the genes encoding OEP64 and TOC64 were significantly up-regulated; however, 16 DEGs was down-regulated and significantly negatively correlated with cinnavalininate (+), but were positively correlated with 5-Hydroxy-l-tryptophan (+), picolinic acid (+), and indoleacetaldehyde (−), and the genes encoding FMO1, AOP1.2, LOC105044482, At4g34880, TYDC, and F3DO2 were significantly down-regulated. In histidine metabolism, four DEGs were up-regulated and negatively correlated with l-Glutamic acid (+) and l-Histidine (+), and the genes involved in amino acid metabolism encoding OAt1g04770 were significantly up-regulated; however, eight DEGs were down-regulated and significantly positively correlated with l-Glutamic acid (+) and l-Histidine (+), and the genes encoding SDC1 and TCTP were significantly down-regulated. In cysteine and methionine metabolism, 55% of DEGs (32 DEGs) were up-regulated and showed a significant positive correlation with S-Adenosylhomocysteine (−), but were significantly negatively correlated with S-Adenosylhomocysteine (+) and glutathione (+), in which the genes involved in amino acid metabolism encoding SAMDC, SAT1, LOC105054414, Os12g0624000, and PCAS-1 were significantly up-regulated. However, 45% of DEGs (26 DEGs) were down-regulated and significantly negatively correlated with S-Adenosylhomocysteine (−), but significantly positively correlated with S-Adenosylhomocysteine (+) and glutathione (+), and the genes encoding ACS3 and HMT1 were significantly down-regulated. In the phenylalanine, tyrosine, and tryptophan biosynthesis pathway, a total of 41 DEGs involved in amino acid metabolism were detected, of which 20% (8 DEGs) were up-regulated and significantly positively correlated with phenylpyruvate (+) and shikimate (+, −), but significantly negatively correlated with 3-Hydroxybenzoate (+) and quinic acid (−). Among them, the genes encoding ASA1 and prudu_014267 were significantly up-regulated. However, 80% of DEGs (33 DEGs) were down-regulated, showing a significant positive correlation with 3-Hydroxybenzoate (+) and quinic acid (−), but were significantly negative correlated with phenylpyruvate and shikimate (+, −), in which the genes encoding TRPA1, ADT6, BAAG/APAT, and DHAPS-1 were significantly down-regulated. In the 2-Oxocarboxylic acid metabolism pathway, a total of 28 DEGs involved in amino acid metabolism were detected, of which 7 DEGs were up-regulated and significantly positively correlated with l-Isoleucine (+), l-Lysine (+), 2-METHYLMALEATE (−), phenylpyruvate (−), and citric acid (−), but significantly negatively correlated with l-Glutamic acid (+), in which the genes encoding CSG, IIL1, and CS (involved in amino acid metabolism) were significantly up-regulated, whereas 21 DEGs were down-regulated and significantly positively correlated with l-Glutamic acid (+), but were significantly negatively correlated with l-Isoleucine (+), l-Lysine (+), 2-METHYLMALEATE (−), phenylpyruvate (−), and citric acid (−), and the genes encoding SSU1, At5g14590, At3g58610, DHAD, and LOC105054976 were significantly down-regulated. In the biosynthesis of amino acids pathway, a total of 169 DEGs involved in amino acid metabolism were detected, 46% of which (78 DEGs) were up-regulated, with significantly positive correlation with shikimate (+, −), l-Isoleucine (+), l-Lysine (+), l-Saccharopine (+), phenylpyruvate (−), S-Adenosylhomocysteine (−), and citric acid (−), but a significant negative correlation with S-Adenosylhomocysteine (+), l-Glutamic acid (+), and l-Histidine (+), and the genes encoding pyruvate kinase (PYK), FBA5, TPIP1, SAT1, RPI2, Os12g0624000, PCAS-1, and CSG were significantly up-regulated; however, 54% of DEGs (91 DEGs) were down-regulated and significantly positively correlated with S-Adenosylhomocysteine (+), l-Glutamic acid (+), and l-Histidine (+), but were significantly negatively correlated with shikimate (+, −), l-Isoleucine (+), l-Lysine (+), l-Saccharopine (+), phenylpyruvate (−), S-Adenosylhomocysteine (−), and citric acid (−), and the genes encoding TRPA1, ADT6, BAAG/APAT, GAPCP2, gpmA1, and DHAPS-1 were significantly down-regulated. Among other amino acid metabolism pathways, in cyanoamino acid metabolism, a total of 23 DEGs involved in other amino acid metabolism pathways were detected, of which 9 were up-regulated, showing a significant positive correlation with l-Isoleucine (+), but showed a significant negative correlation with prunasin (+), in which genes encoding PCAS-1, CYSC, and SHM7 were significantly up-regulated, whereas 14 DEGs were down-regulated and significantly positively correlated with prunasin (+), but significantly negatively correlated with l-Isoleucine (+), in which the genes encoding BGLU12, PCMP-E13, At3g16150, and BGLU22 were significantly down-regulated. In glutathione metabolism, a total of 44 DEGs involved in other amino acid metabolic pathways were detected, of which 24 were up-regulated and showed negative correlation with glutathione (+) and l-Glutamic acid (+), while the genes encoding MGST3, GSTU18, GGCT2-2, PARA, GSTU17, GSTU10, HSP26-A, and GSTU22 were significantly up-regulated; however, 20 DEGs were down-regulated and significantly positively correlated with glutathione (+) and l-Glutamic acid (+), with the genes encoding GSTU19, GOT1, At5g14590, G6PGH1, and RNR1 being significantly down-regulated (Figure 9 and Appendix A, Appendix A).

#### 2.4.2. Carbohydrate, Lipid, and Nucleotide Metabolism Contributes to Cold Stress in Coconut

The results obtained suggest that cold stress has significant effects on carbohydrate, lipid, and nucleotide metabolism (Figure 10 and Appendix A, Appendix A). Starch and sucrose metabolism, carbon metabolism, and galactose metabolism pathways were involved in carbohydrate metabolism. In the starch and sucrose metabolism pathway, a total of 120 DEGs were detected, and 54 were up-regulated and found significantly positively correlated with d-FRUCTOSE (−), but significantly negatively correlated with sucrose (−), in which the GLU3, At1g32860, At2g01630, WAXY, TPP6, and TPPG genes were significantly up-regulated, whereas 66 DEGs were down-regulated and significantly positively correlated with sucrose (−), but significantly negatively correlated with d-FRUCTOSE (−), in which the genes encoding At1g64390, BGLU12, SIT2, TPP6, TPS6, At3g02645, GLU2, HXK3, PDCB5, FLN2, LECRK42, and LECRK41 were significantly down-regulated. In carbon metabolism, 165 DEGs were detected, and 105 DEGs were up-regulated and were significantly positively correlated with citric acid (−) and 2-Dehydro-3-deoxy-d-gluconate (−), but significantly negatively correlated with l-Glutamic acid (+), in which the genes encoding pyruvate kinase (PK), FBA5, FDH1, TPIP1, SAT1, Os06g0486800, and At1g32060 were significantly up-regulated, while 60 DEGs were down-regulated and significantly positively correlated with l-Glutamic acid (+), but significantly negatively correlated with citric acid (−) and 2-Dehydro-3-deoxy-d-gluconate (−), where the genes encoding HXK3, PCMP-E13, GAPCP2, gpmA1, RPI1, At5g14590, G6PGH1, CMDH, OsI_031067, MLTHFR1, EMB3003, LOC105060382, and LOC103721328 were significantly down-regulated. In the galactose metabolism pathway, 31 DEGs were up-regulated and significantly positively correlated with d-FRUCTOSE (−) and raffinose (−), but negatively correlated with sucrose (−), and the genes encoding GOLS2, GOLS1, RFS6, UGEPI48, HXK4, PFK3, LOC109505297, and HXK2 were significantly up-regulated; however, 17 DEGs were down-regulated and had a significant positive correlation with sucrose (−), but had a significant negative correlation with d-FRUCTOSE (−) and raffinose (−), in which the genes encoding Os03g0255100, At3g02645, α-GAL, HXK3, and IVR1 were significantly down-regulated. In the glycerophospholipid metabolism pathway, 50% of DEGs (41 DEGs) were up-regulated and significantly negatively correlated with choline (+) and phosphocholine (+), and the genes involved in lipid metabolism encoding At4g16820, At4g19060, LOC105031993, PAH2, PECT1, SRC2, G3PAT, GDPD1, At1g78690, At1g06800, SYT5, LOC105045290, GPAT2, and GPAT3 were significantly up-regulated. However, 50% of DEGs (41 DEGs) were down-regulated and significantly positively correlated with choline (+) and phosphocholine (+), in which the genes encoding RAM2, OBL1, GPAT3, DGK7, PLMT, GPDH, NPC2, and Os04g0669500 were significantly down-regulated. In the arachidonic acid metabolism pathway, six DEGs were up-regulated and positively correlated with 15-keto-Prostaglandin E2 (+) and delta-12-PGJ2 (+), and the genes involved in lipid metabolism encoding P23-1 and prostaglandin E synthase 2 (PTES2) were significantly up-regulated; however, four DEGs were down-regulated and significantly negatively correlated with 15-keto-Prostaglandin E2 (+) and delta-12-PGJ2 (+), in which genes encoding CYP735A1 and putative epoxide hydrolase (PTEH) were significantly down-regulated (Figure 10 and Appendix A, Appendix A). In the pyrimidine metabolism pathway related to nucleotide metabolism, 41 DEGs were detected, and 9 DEGs were up-regulated and significantly negatively correlated with 5-Methylcytosine (+), deoxycytidine (+, −), pseudouridine (−), and uridine (−); among them, the genes involved in nucleotide metabolism encoding dCTP pyrophosphatase 1(dCTP-PP1), LOC105046338, and DMP3 were significantly up-regulated, whereas 32 DEGs were down-regulated and significantly positively correlated with 5-Methylcytosine (+), deoxycytidine (+, −), pseudouridine (−), and uridine (−), and the genes encoding Os02g0778400, RNR1, CARB, PCMP-H40, UPP, DDB_G0275467, and UMPS1 were significantly down-regulated (Figure 10 and Appendix A, Appendix A).

#### 2.4.3. Biosynthesis of Other Secondary Metabolites and Metabolism of Cofactors and Vitamins Contributes to Cold Tolerance in Coconut

Our study suggests that cold stress had significant effects on the biosynthesis of other secondary metabolites and the metabolism of cofactors and vitamins (Appendix A, Appendix A). In the biosynthesis of other secondary metabolites’ pathway, 16 DEGs involved in the isoquinoline alkaloid biosynthesis (ko00950) pathway were detected; 7 DEGs were up-regulated and significantly negatively correlated with 4-Hydroxycinnamic acid (+), 3,4-Dihydroxybenzaldehyde (+), and berberine (+); and the genes involved in biosynthesis of other secondary metabolites encoding primary amine oxidase (PAO), At1g62810, Prudu_014267, and N4OMT2 were significantly up-regulated, while 9 DEGs were down-regulated and significantly positively correlated with 4-Hydroxycinnamic acid (+), 3,4-Dihydroxybenzaldehyde (+), and berberine (+); the genes encoding AATG/AAT, LOC105044482, TYDC, and polyphenol oxidase (PPOX) were significantly down-regulated. In the flavone and flavonol biosynthesis (ko00944) pathway, three DEGs involved in the biosynthesis of other secondary metabolites encoding 3MAT, UGT73E1, and CYP75B137 were up-regulated and were significantly positively correlated with astragalin (+), apiin (+, −), and apigenin 7-*O*-neohesperidoside (−), but significantly negatively correlated with apigenin 7-*O*-neohesperidoside (+), astragalin (−), and quercitrin (−); however, DEGs encoding 3MAT were down-regulated and positively correlated with apigenin 7-*O*-neohesperidoside (+), astragalin (−), and quercitrin (−), but significantly negatively correlated with astragalin (+), apiin (+, −), and apigenin 7-*O*-neohesperidoside (−). In the flavonoid biosynthesis (ko00941) pathway, 17 DEGs were up-regulated and significantly positively correlated with (−)-Epicatechin (+), daidzin (+), chlorogenic acid (+), neohesperidin (+), and naringin (−), but significantly negatively correlated with Chlorogenate (+), in which genes encoding At5g05600, SALAT, UGT88F3, ANS, UGT88F5, and SAT were significantly up-regulated, whereas 19 DEGs were down-regulated and significantly positively correlated with chlorogenate (+), but were significantly negatively correlated with (−)-Epicatechin (+), daidzin (+), chlorogenic acid (+), neohesperidin (+), and Naringin (−), among which genes encoding At3g11180, At5g05600, LOC105043757, UGT71K1, DFRA, CYP73A4, CYP73A1, C3H, and CER26L were significantly down-regulated. In the phenylpropanoid biosynthesis (ko00940) pathway, 66 DEGs involved in the biosynthesis of other secondary metabolites were detected; 21 DEGs were up-regulated and were significantly positively correlated with chlorogenic Acid (+) and 3,5-Dimethoxy-4-hydroxycinnamic acid (−), but were significantly negatively correlated with 4-Hydroxycinnamic acid (+), coniferol (+), 4-Hydroxy-3-methoxycinnamaldehyde (−), and chlorogenate (−), where the genes encoding SALAT, CAD1, PER64, SAT, CSE, UGT89B1, UGT89B2, ALDH2C4, BAOBT, CYP73A100, and CAD9 were significantly up-regulated, while 45 DEGs were down-regulated and significantly positively correlated with 4-Hydroxycinnamic acid (+), coniferol (+), 4-Hydroxy-3-methoxycinnamaldehyde (−), and chlorogenate (−), but were significantly negatively correlated with chlorogenic acid (+) and 3,5-Dimethoxy-4-hydroxycinnamic acid (−). The genes encoding BGLU12, PER65, CAD6, LOC105043757, GSVIVT00037159001, CYP73A4, LOC103723514, PER12, CYP73A1, and C3H were significantly down-regulated. In the tropane, piperidine, and pyridine alkaloid biosynthesis (ko00960) pathway, six DEGs were up-regulated and were positively correlated with l-Isoleucine (+), l-Lysine (+), and phenylpyruvate (−), and the genes encoding primary amine oxidase (PAO) and Noroxomaritidine/norcraugsodine reductase (NR) were significantly up-regulated; however, seven DEGs were down-regulated and were significantly negatively correlated with l-Isoleucine (+), l-Lysine (+), and phenylpyruvate (−), in which the genes encoding AATG/AAT and SDR3a were significantly down-regulated. In the stilbenoid, diarylheptanoid, and gingerol biosynthesis (ko00945) pathway, six DEGs were up-regulated and were significantly positively correlated with bisdemethoxycurcumin (+) and chlorogenic acid (+), but significantly negatively correlated with chlorogenate (−), in which the genes encoding SALAT, SAT, BAOBT, and CYP73A100 were significantly up-regulated. However, 12 DEGs were down-regulated and significantly positively correlated with chlorogenate (−), but significantly negatively correlated with bisdemethoxycurcumin (+) and chlorogenic acid (+), where the genes encoding LOC105043757, ROMT, CYP73A4, CYP73A1, C3H, and CER26L were significantly down-regulated. In the isoflavonoid biosynthesis(ko00943) pathway, three DEGs encoding CYP81Q32, 3MAT, and CYP81Q32 were up-regulated and were significantly positively correlated with daidzin (+), biochanin A (−), and ononin (−); however, two DEGs encoding CYP81Q32 and 3MAT were down-regulated and significantly negatively correlated with daidzin (+), biochanin A (−), and ononin (−) (Appendix A, Appendix A). In the metabolism of cofactors and vitamins pathway, five DEGs were up-regulated in the vitamin B6 metabolism (ko00750) pathway and were significantly negatively correlated with pyridoxamine 5′-phosphate (+) and pyridoxine (+), in which the genes involved in the metabolism of cofactors and vitamins encoding At1g17710 and THS were significantly up-regulated. However, the one DEG encoding probable pyridoxal 5′-phosphate synthase (PDX2) was down-regulated and significantly positively correlated with pyridoxamine 5′-phosphate (+) and pyridoxine (+). In the biotin metabolism (ko00780) pathway, two DEGs encoding Os08g0327400 and At1g10310 were up-regulated and were significantly positively correlated with 7-Keto-8-aminopelargonic acid (+) and l-Lysine (+). However, eight DEGs were down-regulated and were significantly negatively correlated with 7-Keto-8-aminopelargonic acid (+) and l-Lysine (+), in which the genes encoding BIO3-BIO1, KAS2, and KAS12 were significantly down-regulated. In the ubiquinone and other terpenoid-quinone biosynthesis (ko00130) pathway, 31 DEGs involved in the metabolism of cofactors and vitamins pathway were detected. Seven DEGs were up-regulated and significantly positively correlated with delta-Tocotrienol (+), but significantly negatively correlated with 4-Hydroxycinnamic acid (+), and the genes encoding Prudu_014267, CYP73A100, and COQ5 were significantly up-regulated. Twenty-four DEGs was down-regulated and significantly positively correlated with 4-Hydroxycinnamic acid (+), but significantly negatively correlated with delta-Tocotrienol (+), and the genes encoding At4g27270, CYP73A4, CUMW_146310, CYP73A1, At4g27270, and FQR1 were significantly down-regulated. In nicotinate and nicotinamide metabolism (ko00760), eight DEGs were up-regulated and significantly positively correlated with 6-Hydroxynicotinic acid (+), but significantly negatively correlated with N1-Methyl-2-pyridone-5-carboxamide (+, −) and beta-Nicotinamide d-ribonucleotide (+), and the gene encoding LOC105046338 was significantly up-regulated. However, nine DEGs were down-regulated and significantly positively correlated with N1-Methyl-2-pyridone-5-carboxamide (+, −) and beta-Nicotinamide d-ribonucleotide (+), but significantly negatively correlated with 6-Hydroxynicotinic acid (+), and the genes encoding Os09g0345700, DDB_G0275467, VPDT2, and NAPRT1 were significantly down-regulated. In the folate biosynthesis (ko00790) pathway, 10 DEGs were up-regulated and significantly positively correlated with d-Neopterin (−), but negatively correlated with biopterin (−), where the genes involved in metabolism of cofactors and vitamins encoding RIBA1, MOLC, RIBA1, and FPGS1 were significantly up-regulated; however, 6 DEGs were down-regulated and had a significant positive correlation with biopterin (−), but a significant negative correlation with d-Neopterin (−), in which the genes encoding LOC105052891 and FLACCA were significantly down-regulated (Appendix A, Appendix A).

#### 2.4.4. Translation and Membrane Transport Contributes to Cold Stress in Coconut

Based on the transcriptome and metabolome analysis, cold stress has been found to have a significant impact on translation and membrane transport (Appendix A, Appendix A). In the aminoacyl-tRNA biosynthesis (ko00970) pathway related to translation, 36 DEGs involved in translation pathways were detected, and 13 DEGs were up-regulated and were significantly positively correlated with l-Isoleucine (+) and l-Lysine (+), but significantly negatively correlated with l-Glutamic acid (+) and l-Histidine (+), where the genes involved in translation encoding FAAH, SFH10, Sb10g008780, and phosphatidylinositol transfer protein 3 (PPLTP3) were significantly up-regulated, whereas 23 DEGs were down-regulated and significantly positively correlated with l-Glutamic acid (+) and l-Histidine (+) and significantly negatively correlated with l-Isoleucine (+) and l-Lysine (+). The genes encoding IRL4, DIR19, At5g26710, PTS1, and At3g04600 were significantly down-regulated (Appendix A, Appendix A). In the ABC transporters (ko02010) pathway, 33 DEGs involved in membrane transporters pathway were detected, and 13 DEGs were up-regulated and significantly positively correlated with l-Isoleucine (+), l-Lysine (+), d-FRUCTOSE (−), and raffinose(−), but significantly negatively correlated with glutathione (+), l-Glutamic acid (+), l-Histidine (+), deoxycytidine (+, −), choline (+), maltotriose (−), sucrose (−), deoxyinosine (−), and uridine (−); among them, the genes involved in membrane transport encoding ABCC3, ABCB11, ABCB4, and ABCG36 were significantly up-regulated; however, 20 were down-regulated. On the contrary, they were significantly positively correlated with glutathione (+), l-Glutamic acid (+), l-Histidine (+), deoxycytidine (+, −), choline (+), maltotriose (−), sucrose (−), deoxyinosine (−), and uridine (−), but significantly negatively correlated with l-Isoleucine (+), l-Lysine (+), d-FRUCTOSE (−), and raffinose (−), where the genes encoding ABCB10, ABCB2, ABCG42, ABCG45, ABCB11, ABCG3, ABCB8, ABCG31, ABCB19, ABCG11, and ABCG41 were significantly down-regulated (Appendix A, Appendix A).

## 3. Discussion

Plants resist abiotic stress through various physiological and metabolic regulations [10,32,57]. In this study, we performed physiological, transcriptomic, and metabolomic analyses to uncover the regulatory network of coconut’s response to cold stress. The physiological changes in coconut primarily occurred at lower temperatures; significant changes were observed in metabolite accumulation and gene expression using metabolomics and RNA-Seq analysis.

### 3.1. Physiological Response of Coconut under Cold Stress

Intracellular ROS was believed to play a decisive role in regulating signal transduction events. Abiotic stress led to the accumulation of ROS [63]. Excessive ROS enhances membrane lipid peroxidation and destroys membrane fluidity, resulting in electrolyte leakage (EL) [64], thus destabilizing the plasma membrane of plant cells [65]. Malondialdehyde (MDA) is a lipid peroxide derivative, production of which can aggravate membrane damage [66]. The increase in MDA level indicates the degree of damage to the cell membranes. For example, low-temperature treatment increased the MDA concentration and EL value of yellow horn (*Xanthoceras sorbifolia*), indicating that cold stress may lead to plasma membrane destruction [32]. However, under abiotic stress conditions, ROS homeostasis largely depends on the ROS clearance system [67]. The activity of several antioxidant enzymes (including SOD, CAT, and POD) has been found as a trigger to reduce ROS damage to plant cells, with SOD being an important ROS scavenging enzyme [68]. In this study, we observed an increase in MDA content (Figure 2a), indicating that low temperatures have a considerable effect on coconut seedlings. The activity of reactive oxygen scavenging enzymes (including SOD, CAT, and POD) was reportedly elevated (Figure 2c–e), indicating that they remove ROS produced by cold stress. SOD is a scavenger of peroxide anions, which can disproportionate the anions into H_2_O_2_ and O_2_ [63]. At the same time, POD and CAT catalyze the conversion of H_2_O_2_ to oxygen and water [69]. The increase in SOD activity was found to be lower than that of POD and CAT, which may be due to the higher decomposition ability of POD and CAT to H_2_O_2_ produced by SOD. In conclusion, the enhancement of cold-induced enzyme activity expression may improve the detoxification ability of reactive oxygen species to oxidative stress in coconut. Soluble sugars and proteins are considered to be key osmoregulatory substances in plants, and their accumulation in cytosol can prevent protoplast dehydration and improve the plant’s cold resistance [24,25]. In this study, the contents of soluble sugar and protein reportedly increased significantly under cold stress, reaching a peak on day 20, and decreased gradually from days 20 to 30 (Figure 1c,d), indicating that soluble sugar and protein positively regulate the damage to the coconut cell membrane at low temperatures and reduce it with prolonged cold stress. This decreased content may be attributed to the inhibition of photosynthesis in the late stage of cold stress [16], which substantiates the significant decline in SPAD (Figure 1b). Proline (PRO) is an important amino acid in osmoregulation, which also helps to stabilize subcellular structures, scour free radicals, and buffer cell reductant-oxidant potential under stress conditions [70]. Rapid decompression of PRO after decompression can provide numerous reducing agents, which helps to support mitochondrial oxidative phosphorylation and ATP production. Recovery from stress and repair of stress-induced damage are also important indicators in the study of plant cold resistance [71]. Similar to previous studies, PRO levels found in cold-stressed coconuts were higher than the control (Figure 2b). This suggests that the high level of PRO in coconut can provide a large amount of PRO reducing agents, thus contributing to energy production, which is beneficial for plants to recover from adversity and repair damage caused by stress.

### 3.2. Transcription Factor in Response to Cold Stress in Coconut

Transcription factors play an important role in plant growth and development, and also participate in the plant transcriptional network in response to abiotic stresses [72]. The TF families related to plant stress resistance mainly include AP2/EREBP, MYB, WRKY, and bHLH [73]. Consistent with findings from other plants, our transcriptomic data showed that most differentially expressed TFs belonged to the WRKY, AP2/ERF, HSF, bZIP, MYB, and bHLH families. Previous studies have shown the diversity and complexity of regulatory pathways of TFs in response to cold stress [74]. The AP2/ERF family has been divided into the AP2/ERF, RAV, and DREB subfamilies, which play an important role in multiple stress responses [75]. For example, overexpression of the rice *OsDRE1F* gene can improve the tolerance of *Arabidopsis* and rice to low temperature [76], while the birch *BpERF13* gene can improve the tolerance of *Arabidopsis* and rice to low temperature by up-regulating the CBF gene and reducing the number of ROS, thus demonstrating the positive regulation of cold stress [77]. In this study, AP2/ERF transcription factors were up-regulated in 74% of AP2/ERF TFs and down-regulated in 26% of AP2/ERF TFs under cold stress (Figure 6, Appendix A). These results are consistent with previous reports of cold-stress regulation caused by the presence of the AP2/ERF family [78,79]. WRKY-TFs represented a valuable family for resistance to abiotic stresses, such as cold, heat, and salt. For example, a total of 17 members of the family were induced by refrigeration in japonica rice [78]. In addition, a group of WRKY genes were also found to be cold-regulated in indica rice, and 18 WRKY genes were fully up-regulated in cold-tolerant varieties genotypes [80]. In addition, for the *VbWRKY32* in the seed stage of *Verbena bonariensis,* the transcription level of cold response genes were up-regulated, thereby increasing the activity of antioxidant enzymes and the content of osmoregulatory subunits, thus improving the survival ability under cold stress [81]. In this study, 67% of WRKY TFs were up-regulated and 33% (14 DEGs) were down-regulated after low-temperature treatment (Figure 6, Appendix A). The expressions of 43 WRKY DEGs under cold stress may be helpful to further study the mechanism of their cold resistance. In addition, MYB family members have been shown to be key factors in regulating responses to abiotic stress. Some MYB TFs are involved in the regulation of cold stress. For example, *LcMYB4* expression in *Leymus chinensis* was rapidly induced by cold treatment and actively regulated the cold tolerance of *Arabidopsis* [82]. Similar to previous studies, MYB genes in coconut were induced by the down-regulated and up-regulated mode under cold stress. For example, 38% of MYB TFs were up-regulated and 62% (eight DEGs) were down-regulated (Figure 6, Appendix A). In this study, the transcription of heat shock transcription factors (HSFs) in coconut was induced by cold stress, and 89% of HSF TFs were up-regulated (Figure 6, Appendix A). Some reports have shown that the abundance of HSF TFs (HSFA4A, HSFA6B, HSFA8, and HSFC1) in *Arabidopsis Thaliana* was enhanced by cold stress, while induction was diminished in ice1 mutants, suggesting that HSF was involved in the cold acclimation pathway [83]. In addition, overexpression of HSF TFs can stimulate the synthesis of protective metabolites, such as galactosol-affinity sugars, to improve plant tolerance to abiotic stresses [84]. Therefore, it was proposed that the relatively high transcription level of HSF TFs in coconut under cold stress may be more conducive to its response to cold stress. In addition, bZIP and bHLH TFs have also been reported to be involved in stress response [40,85]. In this study, 80% of bZIP TFs were up-regulated. In the bHLH family, two DEGs encoding BHLH41 (COCN_GLEAN_10013682, COCN_GLEAN_10011130) were significantly up-regulated. However, one DEG encoding BHLH41 (COCN_GLEAN_10013779) was significantly down-regulated (Figure 6, Appendix A). In our results, major TF families, such as AP2/ERF, WRKY, MYB, HSF, bZIP, and bHLH, identified a large number of DEGs that induced expressions under cold stress. Therefore, these TFs have a significant effect on the cold resistance of coconut and may be involved in complex mechanisms related to low-temperature regulation.

### 3.3. The Core Regulator Genes in Response to Cold Stress in Coconut

In many plants, the ICE-CBF-COR signaling pathway is the most important signaling pathway in response to cold stress. This pathway is regulated by CBF/DREB transcription factors to induce low-temperature tolerance [86]. Numerous genetic and molecular analyses have identified C-repeat/DREB binding factors (CBFs) as key transcription factors that play a role in cold domestication and are essential in higher plants [87]. Overexpression of CBF could induce COR expression and improves frost resistance [88]. In this study, the expression of cold-regulated protein, cold-responsive protein kinase, and temperature-induced lipocalin-1 genes were up-regulated under low-temperature stress. A TF ICE1 (SCRM) encoding COCN_GLEAN_10023355 was up-regulated; however, another TF ICE1 (SCRM) encoding COCN_GLEAN_10008268 was down-regulated. Six DREB/CBF transcription factors (DREB1E (COCN_GLEAN_10009450, COCN_GLEAN_10008816, and COCN_GLEAN_10001620), DREB1C, DREB2C, and DREB1G were significantly up-regulated, while two DREB/CBF transcription factors (DREB3 and DREB1F) were significantly down-regulated (Figure 6, Appendix A); in addition, some genes of aquaporins (TIP1-1, PIP2-4, SIP2-1, PIP2-2, NIP1-1, PIP1-2) that were bona fide participants in the cold response at the molecular levels [89,90] were significantly down-regulated, and the gene of aquaporin (SIP1-2) was significantly up-regulated (Appendix A), which not only confirmed the universality of the ICE-CBF-COR signaling pathway in response to low temperatures, but also proved that it is helpful to further study the molecular mechanism of this pathway mediating the cold tolerance of coconut.

Analysis of transcriptome data and KEGG of DEGs showed that many genes are involved in signal perception (MAPK signaling pathway-plant), transduction, and regulation under low-temperature stress. After the plant received the cold signal, the membrane permeability of the Ca^2+^ receptor was enhanced, allowing more Ca^2+^ to enter the plasma, resulting in rapid accumulation of Ca^2+^ in the cytosol. Thus, Ca^2+^ is an important second messenger that plays a key role in response to cold stress [91]. For example, in mesophyll cells of *Arabidopsis*, cold stress can lead to an immediate increase in cytoplasmic free Ca^2+^ concentration and activation of Ca^2+^ permeable channels [92]. Ca^2+^ sensors sense changes in intracellular Ca^2+^ levels through phosphorylation and then transduce signals to turn on downstream signaling processes for cold-specific gene expression, which helps plants adapt to cold stress [93]. Ca^2+^ acts as a secondary messenger in response to cold stress and is recognized by Ca-binding proteins. The main intracellular Ca^2+^ sensors in plants are calmodulin (or calmodulin-binding protein)/calcium-binding protein (or calmodulin-like protein)/calmodulin-binding protein (CAM/CML/CBP), calcium-dependent protein kinases (CPK), and calcineurin B-like proteins/CBL-interacting protein kinases (CBL/CIPK) [9,94]. During this study, in the calcium-binding proteins (CMLs), 85% of DEGs were up-regulated. In calmodulin-related proteins (CAM/CML/CBP), 75% of DEGs were up-regulated. However, in calcium-dependent protein kinases (CPKs), 70% of DEGs were down-regulated. Moreover, in calcineurin B-like protein (CBL), two DEGs encoding CBL9 (CUFF17.251.1) and CBL7 were up-regulated, but three encoding CBL1 (CUFF34.779.2, CUFF5.369.1) and CBL9 (COCN_GLEAN_10019851) were down-regulated. In CBL-interacting protein kinase (CIPK), 75% of DEGs were up-regulated (Figure 6, Appendix A). Similarly, some studies also suggested that the primary genes involved in intracellular Ca^2+^ sensors are involved in Ca^2+^ signaling and promote an increase in intracellular Ca^2+^ concentration to activate various transcriptional cascades and potential candidate genes that activate downstream COR gene expression [95].

The MAPK cascade pathway is an important signal transduction pathway involved in abiotic stress response; it plays an important role in cold reactions [96]. The MAPK cascade pathway is composed of three parts: MAP kinase (MAPK/MPK), MAP kinase kinase (MAPKK/MKK), and MAP kinase kinase kinase (MAPKKK/MEKK) [97]. According to KEGG pathway annotation, 51% of DEGs were up-regulated and 49% were down-regulated in the MAPK signaling pathway-plant pathway (Figure 5, Appendix A). It has been reported that the MAPKKK (MAP kinase kinase kinase)–MAPKK (MAP kinase kinase)-MAPK cascade of *Arabidopsis thaliana* can be activated by ROS, and CRLK1 can interact with MEKK1. Phosphorylation activates the MEKK1-mediated MAPK pathway (MEKK1-MKK2-MPK4/6), thereby enhancing the plant’s cold tolerance [98,99]. For MAPK in this study, five DEGs encoding MPK5, MPK14, MPK3, MPK1 (COCN_GLEAN_10018606, COCN_GLEAN_10018607), and CBL7 were up-regulated, and MPK5 and MPK14 were significantly up-regulated; however, MPK9 (COCN_GLEAN_10007573, COCN_GLEAN_10003541, COCN_GLEAN_10003954) and MPK10 were down-regulated. In MAPKK/MKK, two DEGs encoding MKK1 and MKK2 were up-regulated, and MKK1 was significantly up-regulated; however, six DEGs encoding MKK5, MKK9, MKK3, and MKK4 (COCN_GLEAN_10020255, COCN_GLEAN_10022613, COCN_GLEAN_10005503) were down-regulated, and MKK5 was significantly down-regulated. In MAPKKK/MEKK, 12 DEGs were up-regulated, and 8 DEGs encoding MAPKKK18 (COCN_GLEAN_10015847, COCN_GLEAN_10018889), MAPKKK3 (COCN_GLEAN_10000563, CUFF3.335.1), and MEKK1 (Cocos_nucifera_newGene_19341, COCN_GLEAN_10013851, COCN_GLEAN_10003127, COCN_GLEAN_10024549) were significantly up-regulated; however, only 2 DEGs encoding MEKK(NPK1) and MEKK(YDA) were down-regulated (Figure 6, Appendix A). Similar to other previous plant studies, the activation and expression of defense genes in coconut may be mediated by the interaction of the MAPK cascade signaling pathway under cold stress, which can improve the cold tolerance of coconut seedlings.

Endogenous hormones and their signaling pathways play a key role in regulating plant defense mechanisms against various biological stresses [100,101]. ABA accumulation is a key event in abiotic stress response [102]. ABA had been reported to accumulate in many species, including wheat, pepper, and bananas, all of which exhibit increased expression of ABA biosynthesis pathway genes under low-temperature stress [9,103,104]. Abiotic stress induces ABA accumulation and combines with PYP/PYL receptors to inhibit PP2C activity, thereby activating protein kinases (SnRK2s) and causing them to phosphorylate downstream transcription factors [105,106]. In this study, ABA content was significantly increased under cold stress (Figure 2f). Most ABA signaling pathway genes encoding PYL3, PYL8, PP2C, RK2, and DPBF3 were up-regulated; however, a few ABA signaling pathway genes encoding PYL10, ABI5, DPBF3, and PYL1 were significantly down-regulated (Appendix A), suggesting that cold-treatment-induced endogenous ABA accumulation directly triggers the expression of some important ABA signaling pathway genes. This further promoted the transcription of the COR gene in response to coconut cold resistance.

Auxin is an endogenous small molecule that works synergistically with other hormonal pathways and has a significant effect on plant growth and stress response [107,108]. Auxin response factor and auxin-responsive protein are key components of the auxin signaling pathway and are regulators of plant growth and stress response [109]. The auxin signal of strawberry was severely blocked under low temperature, indicating that it plays a very important role in cold stress [110]. In this study, under cold stress, IAA content increased significantly, reaching a peak at day 20, slowly decreasing on day 30, and then stabilizing (Figure 2g). In auxin-responsive proteins and the factors, 24 DEGs including the auxin-responsive proteins (SAUR32, ARGOS, SAUR71, Os09g0247700, IAA10, SAUR36) and the auxin response factors (ARF1, ARF8, ARF12, ARF6, ARF9, ARF18) were significantly up-regulated (*p* < 0.05). However, 21 DEGs were down-regulated, in which the auxin-responsive proteins (IAA10, IAA16, IAA6, SAUR71, PIN1C, PIN3A, AUX28, LAX2, SAUR50, SAUR36) and the auxin response factors (ARF19, ARF17, ARF2) were significantly down-regulated (Appendix A). These genes, which were related to auxin-responsive proteins and factors, and the auxin-responsive proteins were induced under cold stress. Therefore, the auxin signaling pathway and its related genes played a very important role in response to cold stress in coconut.

### 3.4. The Major Significant Enriched KEGG Pathways in Transcription and Metabolism in Response to Cold Stress in Coconut

Low temperatures resulted in significant differences in gene and metabolite expression patterns in coconut seedlings. Large amounts of DEGs were detected from coconut seedlings exposed to low temperatures (Appendix A). According to KEGG enrichment analysis, most DEGs-enriched KEGG pathways were “plant hormone signal transduction”, “MAPK signaling pathway-plant”, “plant-pathogen interaction”, “biosynthesis of amino acids”, “amino sugar and nucleotide sugar metabolism”, “glycerophospholipid metabolism”, “circadian rhythm-plant”, “carbon metabolism”, “starch and sucrose metabolism”, “glycolysis/Gluconeogenesis”, “purine metabolism”, “phenylpropanoid biosynthesis”, “steroid biosynthesis”, and “lipoic acid metabolism” (Figure 5, Appendix A). Similar results were found in the cold-stress response of *Xanthoceras sorbifolia* and *Brassica napus* [32,53].

Many DAMs, including phenylpropanoids and polyketides, lipids and lipid-like molecules, benzenoids, organoheterocyclic compounds, organic oxygen compounds, organic acids and derivatives, nucleosides, nucleotides, and analogues, were identified from coconut seedlings exposed to low temperatures (Figure 7d,h, Appendix A). Similar results were found in cold-stress responses of pepper and peanut [44,55]. In positive- and negative-ion mode, most DAM co-enriched KEGG pathways were “amino acid metabolism”, “biosynthesis of other secondary metabolites”, “membrane transport”, and “metabolism of cofactors and vitamins”. In addition, the positive-ion DAMs significantly enriched KEGG pathway included “metabolism of other amino acids”, “lipid metabolism”, and “translation”, while the negative-ion DAMs significantly enriched KEGG pathways contained “Carbohydrate metabolism” and “Nucleotide metabolism” (Figure 7c,g and Appendix A, Appendix A). The KEGG enrichment pathways of these DAMs have been confirmed in other plants in response to cold stress [9,111].

### 3.5. Combined Analysis of Transcription and Metabolism in Response to Cold Stress in Coconut

According to the combination analysis of transcription and metabolism in response to cold stress in coconut, KEGG enrichment analysis showed that DEGs and DAMs in positive- and negative-ions modes were significantly enriched in the KEGG pathways, including “amino acid metabolism”, “metabolism of other amino acids”, “biosynthesis of other secondary metabolites”, “metabolism of cofactors and vitamins”, “carbohydrate metabolism”, “lipid metabolism”, “nucleotide metabolism”, “translation”, and “membrane transport”(Figure 8, Appendix A).

#### 3.5.1. Amino Acid Metabolism in Response to Cold Stress in Coconut

Previous reports have shown that various stress response metabolites in plants were synthesized by the amino acid metabolism pathway, suggesting that amino acid metabolism played an important role in plant response to stress [112]. In this study, many genes and metabolites were induced from the “amino acid metabolism” pathway, including “lysine biosynthesis”, “tryptophan metabolism”, “cysteine and methionine metabolism”, “lysine degradation”, “histidine metabolism”, “phenylalanine, tyrosine and tryptophan biosynthesis”, “tyrosine metabolism”, “2-Oxocarboxylic acid metabolism”, “biosynthesis of amino acids”, “cyanoamino acid metabolism”, and “glutathione metabolism”. Moreover, the DEGs and DAMs of these pathways were significantly correlated (*p* < 0.05) (Figure 9, Appendix A). For example, in the lysine biosynthesis pathway, the seven genes encoding AGD2, LOC105058024, COCN_GLEAN_10015782, DHDPS2, AK1, DAPF, and DHDPS2 were down-regulated and were significantly positively correlated with the down-regulated l-2-Aminoadipate adenylate (+), while being significantly negatively correlated with the up-regulated metabolites, such as l-Lysine (+) and l-Saccharopine (+); however, on the contrary, it was in terms of the up-regulated LYSA gene correlation with metabolites. In lysine degradation (ko00310), 15 DEGs were up-regulated and significantly positively correlated with l-Lysine (+) and l-Saccharopine (+), while 14 DEGs were down-regulated. In the biosynthesis of amino acids (ko01230) pathway, a total of 169 DEGs were identified to be involved in amino acid metabolism; 46% of DEGs were up-regulated and were significantly positively correlated with shikimate (+, −), l-Isoleucine (+), l-Lysine (+), l-Saccharopine (+), phenylpyruvate (−), S-Adenosylhomocysteine (−), and citric acid (−), but significantly negatively correlated with S-Adenosylhomocysteine (+), l-Glutamic acid (+), and l-Histidine (+); however, 54% of DEGs were down-regulated (Figure 9 and Appendix A, Appendix A). The study showed that the genes involved in the amino acid metabolic pathway were up-regulated or down-regulated in response to cold response, which was closely related to or consistent with the metabolites of this pathway. In short, the changes observed at both the transcriptome and metabolic levels strongly suggest that accumulation of these cold response metabolites via synthetic or metabolic pathways may contribute to cold-stress defense, thereby minimizing cold damage.

#### 3.5.2. Carbohydrate Metabolism in Response to Cold Stress in Coconut

Carbohydrates are the main products of photosynthesis. They are the nutrients and important regulators required for plant growth, energy metabolism, and stress response [21]. Soluble sugars are known to function not only as osmoprotectants [113,114], but also to provide cold resistance in plants as a ROS scavenger [115]. In this study, carbohydrate metabolism mainly included starch and sucrose, carbon, and galactose metabolisms. Metabolome analysis showed that d-FRUCTOSE (−), citric acid (−), 2-Dehydro-3-deoxy-d-gluconate (−), and raffinose were up-regulated; however, sucrose (−) and l-Glutamic acid (+) were down-regulated. In the starch and sucrose metabolism pathway, 54 DEGs were up-regulated and significantly positively correlated with d-FRUCTOSE (−) (*p* < 0.05), but were significantly negatively correlated with sucrose (−); however, 66 DEGs were down-regulated (Figure 10 and Appendix A, Appendix A). Similarly, previous studies have also confirmed that most DEGs and DAMs are mainly enriched in different pathways involved in carbohydrate metabolism, among which starch and sucrose metabolism were significantly enriched, which played a crucial role in the adaptation of *Brassica napus* to cold stress [53]. In carbon metabolism, 105 DEGs were up-regulated and significantly positively correlated with citric acid (−) and 2-Dehydro-3-deoxy-d-gluconate (−), but significantly negatively correlated with l-Glutamic acid (+); however, 60 DEGs were down-regulated. In the galactose metabolism pathway, 31 DEGs were up-regulated and significantly positively correlated with d-FRUCTOSE (−) and raffinose (−), but significantly negatively correlated with sucrose (−), whereas 17 DEGs were down-regulated (Figure 10 and Appendix A, Appendix A). Several studies have highlighted the role of these genes in cold stress [113,116]. For example, overexpression of galactinol synthase in cold-induced wheat increases the levels of galactinol and raffinose and confers a higher tolerance to low-temperature stress [117]. Similarly, results obtained also suggest that genes and metabolites involved in carbohydrate metabolism contribute significantly to cold resistance in coconuts.

#### 3.5.3. Flavonoid Metabolism in Response to Cold Stress in Coconut

A subset of plant-specific accumulative metabolites, and the genes involved in secondary metabolism, including flavonoid biosynthesis, flavone and flavonol biosynthesis, and phenylpropanoid biosynthesis, may play a key role in plant response to cold stress [117]. In this study, metabolome detection and analysis identified many flavonoid and secondary metabolites (Appendix A). In the flavone and flavonol biosynthesis pathway, three DEGs encoding 3MAT, UGT73E1, and CYP75B137 were up-regulated and were significantly positively correlated with astragalin (+), apiin (+, −), and apigenin 7-*O*-neohesperidoside (−), but significantly negatively correlated with apigenin 7-*O*-neohesperidoside (+), astragalin (−), and quercitrin (−); however, DEGs encoding 3MAT was down-regulated. In the flavonoid biosynthesis pathway, 17 DEGs were up-regulated and were significantly positively correlated with (−)-Epicatechin (+), daidzin (+), chlorogenic acid (+), neohesperidin (+), and naringin (−), but significantly negatively correlated with chlorogenate (+), whereas 19 DEGs were down-regulated (Appendix A, Appendix A). It can be seen that cold stress in coconut seedlings has significant effects on flavone and flavonol biosynthesis and flavonoid biosynthesis genes, and related metabolites. Previous studies have also shown that the gene expression of flavonoid biosynthesis is closely related to enhancement of low-temperature tolerance [118]. Flavonoids also played an important role in alleviating cold-stress responses by eliminating ROS produced by cold stress [119]. Flavonoids rich accumulation can reduce cold-stress damage [55,111]. Therefore, it can be inferred that flavone and flavonol biosynthesis and flavonoid biosynthesis genes and related metabolites are very important for the cold resistance of coconut seedlings. Phenylpropanoid biosynthesis is one of the most important metabolites in plants, producing a large number of secondary metabolites, such as lignin and flavonoids [120]. Moreover, in the phenylpropanoid biosynthesis pathway, 66 DEGs were detected, involved in the biosynthesis of other secondary metabolites; 21 were up-regulated and were significantly positively correlated with chlorogenic acid (+) and 3,5-Dimethoxy-4-hydroxycinnamic acid (−), but significantly negatively correlated with 4-Hydroxycinnamic acid (+), coniferol (+), 4-Hydroxy-3-methoxycinnamaldehyde (−), and chlorogenate (−), while 45 DEGs were down-regulated (Appendix A, Appendix A). Recent studies have also shown that cold stress induced the expression of structural genes in the phenylpropanoid pathway, including chalcone synthetase (CHS) and 4-coumarin-CoA ligase (4CL). Flavonoids and lignin in loquat (*Eriobotrya japonica*) [121] and apple (*M. domestica*) [122] have promoted the adaptation to the low-temperature environment. In this study, it was also found that cold stress induced the expressions of these genes and associated metabolites in the phenylpropanoid biosynthesis pathway of coconut seedlings (Appendix A, Appendix A). Therefore, the development of gene and metabolite differences in phenylpropanoid biosynthesis was a positive response to cold stress in coconut seedlings. These findings suggest that advances in genetic and metabolic pathways are highly effective in reducing cold-stress damage. At the same time, the aggregation of several genes involved in the metabolism and synthesis of amino acids, sugars, and flavonoids and specific cold-reaction metabolites revealed the cold-stress response and helped to elucidate the molecular mechanism of cold-stress tolerance.

#### 3.5.4. Lipid Metabolism in Response to Cold Stress in Coconut

Lipid metabolism is a biomarker of lipid damages in the plant’s response to cold stress [15]. For the glycerophospholipid metabolism pathway in this study, 50% of DEGs were up-regulated and significantly negatively correlated with choline (+) and phosphocholine (+); however, 50% of DEGs were down-regulated. In the arachidonic acid metabolism pathway, six DEGs was up-regulated and significantly positively correlated with 15-keto-Prostaglandin E2 (+) and delta-12-PGJ2 (+), whereas four DEGs were down-regulated (Figure 10, Appendix A). This study demonstrated that the expression of genes and metabolites in glycerophospholipid metabolism and arachidonic acid metabolism pathways involved in the lipid metabolism of coconut were induced by cold stress. Moreover, significant correlation has been identified between their DEGs and DAMs, which suggested that lipid metabolism was a very important metabolic pathway in the active response of coconut cold stress.

#### 3.5.5. Nucleotide Metabolism in Response to Cold Stress in Coconut

Nucleotide metabolism can be attributed to cold-induced damage. The study found that nucleotide metabolites, such as cyclic AMP (adenosine monophosphate), adenosine, 2-hydroxyadenosine, cytidine, adenine, β-nicotinamide mononucleotide, guanosine monophosphate, guanosine, and 2-hydroxy-6-aminopurine, increased when *A. arguta* was exposed to cold, as cold stress can lead to nucleotide damage and stimulate the increase of nucleotide metabolism [15]. For the pyrimidine metabolism pathway in this study, related to nucleotide metabolism, 9 were up-regulated and significantly negatively correlated with 5-Methylcytosine (+), deoxycytidine (+, −), pseudouridine (−), and uridine (−); however, 32 DEGs were down-regulated (Figure 10, Appendix A). The results showed that cold-stressed coconut seedlings induced the expression of genes and associated metabolites in the pyrimidine metabolism pathway involved in nucleotide metabolism, suggesting that nucleotide metabolism was very important for the positive response to cold stress in coconut.

## 4. Materials and Methods

### 4.1. Plant Materials and Cold Treatments

The coconut fruits (Yellow dwarf coconut) (Wenye No. 2) were grown at the Institute of Coconut Research, Chinese Academy of Tropical Agricultural Sciences (Haikou, China). They were planted in plastic culture bags (25 cm in diameter and 35 cm in depth) filled with a mixture of soil, organic fertilizer, and coconut bran (10:2:1). The plants were grown in a culture room for 2 months, with a relative humidity of 80–90%, a photoperiod of 16 h/8 h (light/dark), and a temperature cycle of 25 °C/25 °C (day/night). Next, the highly consistent coconut seedlings (60 cm) were selected and divided into two groups for the treatment experiment. One group (20 plants) was stored in the greenhouse for 30 d. The coconut leaves were harvested at 0 (CK_0_), 10 (CK_10_), 20 (CK_20_), and 30 d (CK_30_). The other group of coconut seedlings (15 plants) were transferred to the controlled greenhouse for cold treatment at 5 °C, with the same photoperiod. The coconut leaves were harvested at 0 (LT_0_, represented by CK_0_ samples; thus, no collection was required), 10 (LT_10_), 20 (LT_20_), and 30 d (LT_30_). Leaves treated with CK_0_, CK_10_, CK_20_, CK_30_, LT_10_, LT_20_, and LT_30_ were selected for growth index and physiological index analysis; in addition, leaves treated with CK_30_ and LT_30_ were subjected to transcriptome sequencing and metabolite analysis, with each treatment being conducted in three biological replicates. At the end of each sample collection, the leaves were immediately frozen in liquid nitrogen and stored at −80 °C until further analysis.

### 4.2. Measurements of Plant Height, Dry Weight, and Soil and SPAD Values

Plant height (accuracy: 1 mm) was measured using a tape measure. The SPAD value of coconut leaves was measured using the SPAD chlorophyll analyzer (SPAD-502 Plus, Konica Minolta, Tokyo, Japan). To assess the dry weight of coconut seedlings, fresh stems, leaves, and roots of coconut seedlings were dried at 105 °C for 15 min and 70 °C for 72 h, and then weighed using an electronic balance (Labpro, Shanghai, China). The dry weight of the whole plant (stem + leaf + root) was calculated.

### 4.3. Physiological Index Measurements

Coconut seedlings were treated at 25 °C and 5 °C for 0 (CK), 10 (CK_10_, LT_10_), 20 (CK_20_, LT_20_), and 30 days (CK_30_, LT_30_), and leaves of each treatment were collected for physiological analysis, all of which were performed with three biological replicates. ABA, IAA, ZR, GA, SS, SP, Pro, and MDA contents and SOD, POD, and CAT activities were determined. First, 0.1000 g coconut leaf sample was accurately weighed and was mixed with pre-cooled PBS in a ratio of 1:10 by weight (g) to volume (mL). These mixed samples were ground at high speed and centrifuged at 2500 rpm for 10 min. Then, 50 µL of supernatant was measured for determination. The IAA, GA, ABA, ZR, MDA, SP, Pro, SOD (A001-3-2), CAT(A007-1-1), and POD (A084-3-1) kits and standards were obtained from the Nanjing Institute of Biological Engineering, and the measurements were carried out in strict accordance with the manufacturer’s instructions and according to the methods reported by Li (2000) [123]. We used a 1 cm light path colorimetric tube and a blank colorimetric tube to set the baseline. Absorbance was measured with an enzyme labeling meter (DG5033A, Nanjing Huadong Electronics Group Medical Equipment, Nanjing, China). Wavelengths were set to 595 nm (SP), 620 nm (SS), 532 nm (MDA), 520 nm (Pro), 550 nm (SOD), 405 nm (CAT), 420 nm (POD), and 450 nm (IAA, ABA, GA, ZR). All measurements were taken within 10 min of adding the termination solution. The concentration/activity was calculated from the absorbance value combined with the formula for each physiological indicator.

### 4.4. RNA Extraction and RNA-Sequencing (RNA-Seq)

Total RNA was extracted from frozen samples using the improved cetyltrimethyl ammonium bromide (CTAB) method. The purity and integrity of RNA were evaluated visually by agarose gel electrophoresis. RNA concentrations were measured using a NanoDrop 2000 spectrophotometer (Thermo Fisher Scientific, Waltham, MA, USA). The Agilent 2100 Bioanalyzer system (Agilent Technologies, Palo Alto, CA, USA) was used to quantify the integrity of RNA. Library assembly and RNA-seq analysis were performed at Beijing Biomarker Biotechnology Company (Beijing, China) and Beijing Biomarker Cloud Technology Company (Beijing, China) [124]. Sequencing was performed by using the NEBNext^®^ Ultra™ II RNA Library Prep Kit (New England Biolabs, Ipswich, MA, USA) and adding an index code to each sample. These libraries were sequenced on the Illumina^®^ HiSeq2500 platform (Illumina, San Diego, CA, USA). Each sample was sequenced thrice. Raw reads were filtered by removing low-quality reads and adapters. Using the transcript splicing Alignment Hierarchical Index (HISAT 2) program [125], clean sequences were located to the reference coconut genome (http://creativecommons.org/licenses/by/4.0/, accessed on 1 August 2022) [126]. Gene functions were annotated using the following databases: NCBI Non-redundant Protein Sequence (Nr), Homologous Protein Cluster (COG/KOG), Swiss-PROT protein sequence Database, Kyoto Encyclopedia of Genes and Genomes (KEGG), Homologous Protein Families (Pfam), and Gene Ontology (GO) [127,128]. To verify transcriptional expression levels for all samples of each transcription region, RESM software (3.8.6) was used to calculate the fragments per kilobase of transcript per million fragments mapped (FPKM) [129]. DESeq software (1.6.3) was used to analyze the differential gene expression among the samples, and the Benjamini–Hochberg method was used to determine its significance. The definition of DEGs is based on |fold change (FC)| ≥ 2 and FDR < 0.01(log_2_|fold change (FC)| ≥ 1). The False Discovery Rate (FDR) was obtained by correcting the difference significance *p*-value [130]. The GOseq R software package (2.18.0) was used for GO enrichment analysis of DEGs [131,132,133]. KEGG pathway enrichment analysis of DEGs was performed using KEGG Orthology Based Annotation System (KOBAS) software (3.0) [134].

### 4.5. Metabolite Analysis

Sample preparation, metabolomics, and data analysis was conducted by Beijing Biomarkers Biological Technology Co., Ltd. (Beijing, China) (http://www.biomarker.com.cn/, accessed on 15 August 2022). The frozen coconut leaves were ground into powder in liquid nitrogen, and 100 mg of the powder was added to a 1.5 mL Eppendorf tube. Then, 1.0 mL 70% methanol solution was extracted at 4 °C for 24 h and centrifuged at 10,000× *g* and 4 °C for 10 min. The extract was filtered through a 0.22 µm nylon membrane and analyzed by liquid chromatography–mass spectrometry (LC-MS). Extract from each treatment and mixture of three duplicate samples was used to prepare quality control samples. During the analysis, each quality control sample was measured together with the corresponding three experimental samples to check the stability of the analysis conditions. The metabolites of leaf extract (10 µL) were analyzed by ultra-performance LC-electrospray ionization mass spectrometry (UPLC-ESI-MS/MS). Aqueous chromatographic separation was performed at 40 °C using a UPLC HSS T3 C18 column (2.1 mm × 100 mm, i.e., 1.8 µm) (Waters, Milford, MA, USA). The mobile phase consisted of water containing 0.04% acetic acid (mobile phase A) and acetonitrile containing 0.04% acetic acid (mobile phase B). The linear gradient procedure for elution was set to 0–11.0 min from 5% to 95% B, 11.0–12.0 min from 95% to 5%, and 12.0–15.0 min from 5% to 5%. The flow rate of the mobile phase was 0.40 mL/min. The API 4500 QTRAP LC-MS/MS system (AB SCIEX, Framingham, MA, USA) was used for LC-MS/MS analysis. ESI source parameters were as follows: turbo spray, ion source, source temperature 550 °C, ion spray voltage 5.5 kV. The curtain gas pressure was 25 pounds per square inch (psi), the ion source gas I pressure was 55 psi, and the gas II pressure was 60 psi. A multi-reaction monitoring experiment was carried out with 5 psi nitrogen colliding gas, and the results of quadrupole scanning were obtained.

The identification of metabolites was based on the public database and cloud technology database of Beijing Biomarker Biotechnology Co., Ltd. (Beijing, China) [124]. The raw data collected using MassLynx V4.2 was processed using Progenesis QI 3.0 software for peak extraction, peak alignment, and other data processing operations; the data were identified based on Progenesis QI software, the online METLIN (2019) database, and the self-built database of Baimaike (http://www.biomarker.com.cn/, accessed on 1 September 2022); and theoretical fragment identification was performed. The quality deviation was within 100 ppm. The structures of the metabolites were analyzed using standard metabolic procedures. The metabolites were quantitatively analyzed by the multiple reaction monitoring method. All the identified metabolites were analyzed by partial least squares discriminant analysis (PLS-DA). Principal component analysis (PCA) and orthogonal PLS-DA (OPLS-DA) were used to identify potential biomarkers. For the choice of biomarkers, with projected variable importance (VIP) ≥ 1, folding change (FC) ≥ 2 (up-regulated), or ≤0.5 (down-regulated), *p* < 0.05 was used as the criterion for screening significant differentially accumulated metabolites (DAMs).

### 4.6. Integrated Metabolome and Transcriptome Analyses

Pearson correlation tests were performed to determine the correlation between DEGs and DAMs. The Pearson correlation coefficient (PCCs) between DEGs and DAMs was calculated using the “corrplot” in the R package. Genes and metabolites with PCC > |0.8| and *p* < 0.05 were used to establish correlation heat maps and networks. In the correlation analysis, we compared the KEGG pathway where DEGs and DAMs were enriched (*p* < 0.05). DEGs and DAMs with high correlation in KEGG pathways (*p* < 0.05) were selected for analysis and network mapping [135,136,137].

### 4.7. Quantitative Real-Time PCR (qRT-PCR) Analysis

The DEGs of coconut seedlings identified by RNA-seq were verified by qRT-PCR. A gene-specific RT-qPCR primer (Appendix A) was designed. The qPCR was run on the Photocycling^®^ 480II real-time system (Roche, Carlsbad, CA, USA) on a 96-well plate, using HiefeqPCR SYBR green main mixture (Not-Rox) (Yeasen Biotech, Shanghai, China) according to manufacturer’s instructions. The thermal cycle steps consisted of denaturation at 95 °C for 5 min, followed by 40 cycles at 95 °C for 10 s and 60 °C for 30 s. All qRT-PCR analyses were performed using three bio-replicates, each of which contained three technical replicates. The internal reference gene (β-actin) was used for normalization. The 2^−∆∆CT^ method was used to calculate the expression levels of differential target genes in the reference control group [138].

### 4.8. Statistical Analysis

Data were analyzed by Excel (2010) (Microsoft Corporation, Redmond, WA, USA) and expressed as mean ± standard deviation (SD). For statistical analysis, SPSS software was used (Version 20.0; SPSS, Chicago, IL, USA) to conduct one-way analysis of variance (ANOVA) and Student’s *t*-test, and the results were used to determine the difference and statistical significance at *p* < 0.05.

## 5. Conclusions

We investigated the response of coconuts to cold stress at the physiological, transcriptional, and metabolomic levels. Our results suggest that coconuts reduce cold damage by enhancing CAT, POD, and SOD activity and increasing the accumulation of soluble sugars and proteins in cells. The MAPK signaling pathway-plant, plant hormone signal transduction, plant–pathogen interaction, biosynthesis of amino acids, amino sugar and nucleotide sugar metabolism, carbon metabolism, and starch and sucrose metabolism pathways played an important role in the response of coconut to cold stress. TFs, including WRKY, AP2/ERF, HSF, bZIP, MYB, and bHLH, potentially played a critical role in regulating cold-stress-related genes to improve cold resistance in coconut. These results provide a comprehensive understanding of the mechanism of the low-temperature stress response in coconut and guidance for the development of cold-tolerant coconut varieties by identifying candidate genes and incorporating them into molecular breeding programs. This study also deepens the understanding of the complex regulatory mechanisms in plants subjected to cold stress. However, further research should explore the functions of differential genes and metabolites involved in these key regulatory pathways to further understand the mechanisms of the coconut’s low-temperature response.

## Figures and Tables

**Figure 1 ijms-24-14563-f001:**
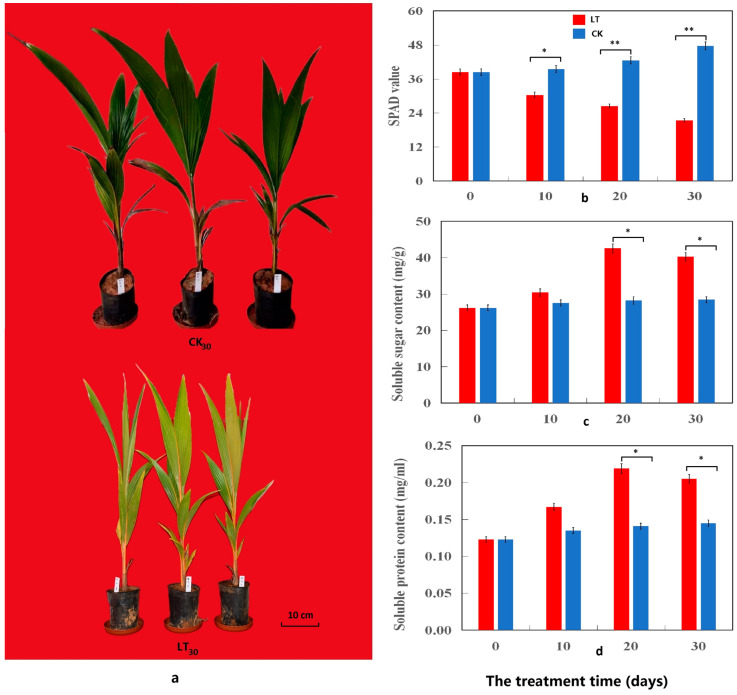
Different physiological responses of coconut seedlings to cold stress. (**a**) Experimental photos of coconut seedlings. (**b**) Changes in SPAD of coconut at 0, 10, 20, and 30 days in LT and CK. (**c**,**d**) SP and SS contents of coconut at 0, 10, 20, and 30 days in LT and CK. The error bar represents SD of the mean of the three biological replicates. * asterisks represent significant differences between the two treatments determined by Student’s *t*-test (* *p* < 0.05; ** *p* < 0.01).

**Figure 2 ijms-24-14563-f002:**
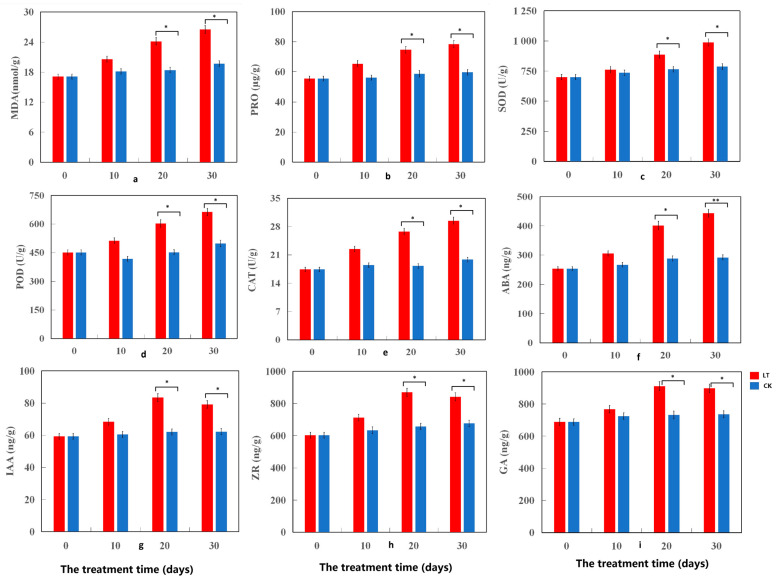
Different antioxidant enzymes and endogenous hormone responses of coconut seedlings to cold stress. (**a**,**b**) MDA and PRO of coconut seedlings at 0, 10, 20, and 30 days in LT (red legend) and CK (blue legend). (**c**–**e**) The SOD, POD, and CAT activities of coconut seedlings at 0, 10, 20, and 30 days in LT and CK. (**f**–**i**) The ABA, IAA, ZR, and GA contents of coconut seedlings at 0, 10, 20, and 30 days in LT and CK. * asterisk indicates a significant difference between the two treatments determined by Student’s *t*-test (* *p* < 0.05; ** *p* < 0.01).

**Figure 3 ijms-24-14563-f003:**
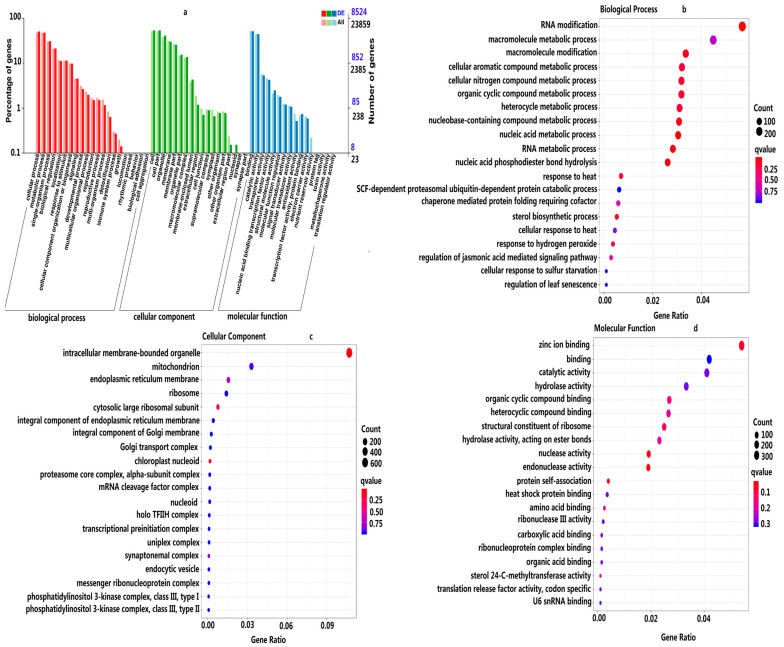
GO classification of differential expression genes (DEGs) in LT_30_ vs. CK_30_. (**a**) GO annotation classification statistical map of DEGs. The *X*-axis is the GO classification, the left side of the *Y*-axis is the percentage of the number of genes, and the right side is the number of genes. (**b**) GO rich distribution point diagram of DEGs in biological process. (**c**) GO rich distribution point diagram of DEGs in cellular component. (**d**) GO rich distribution point diagram of DEGs in molecular function. The *X*-axis is the gene ratio, that is, the proportion of the genes of interest in the entry to the number of DEGs, and the *Y*-axis is each GO entry. The size of the dots represents the number of DEGs annotated in the pathway, and the color of the dots represents the q value of the hypergeometric test.

**Figure 4 ijms-24-14563-f004:**
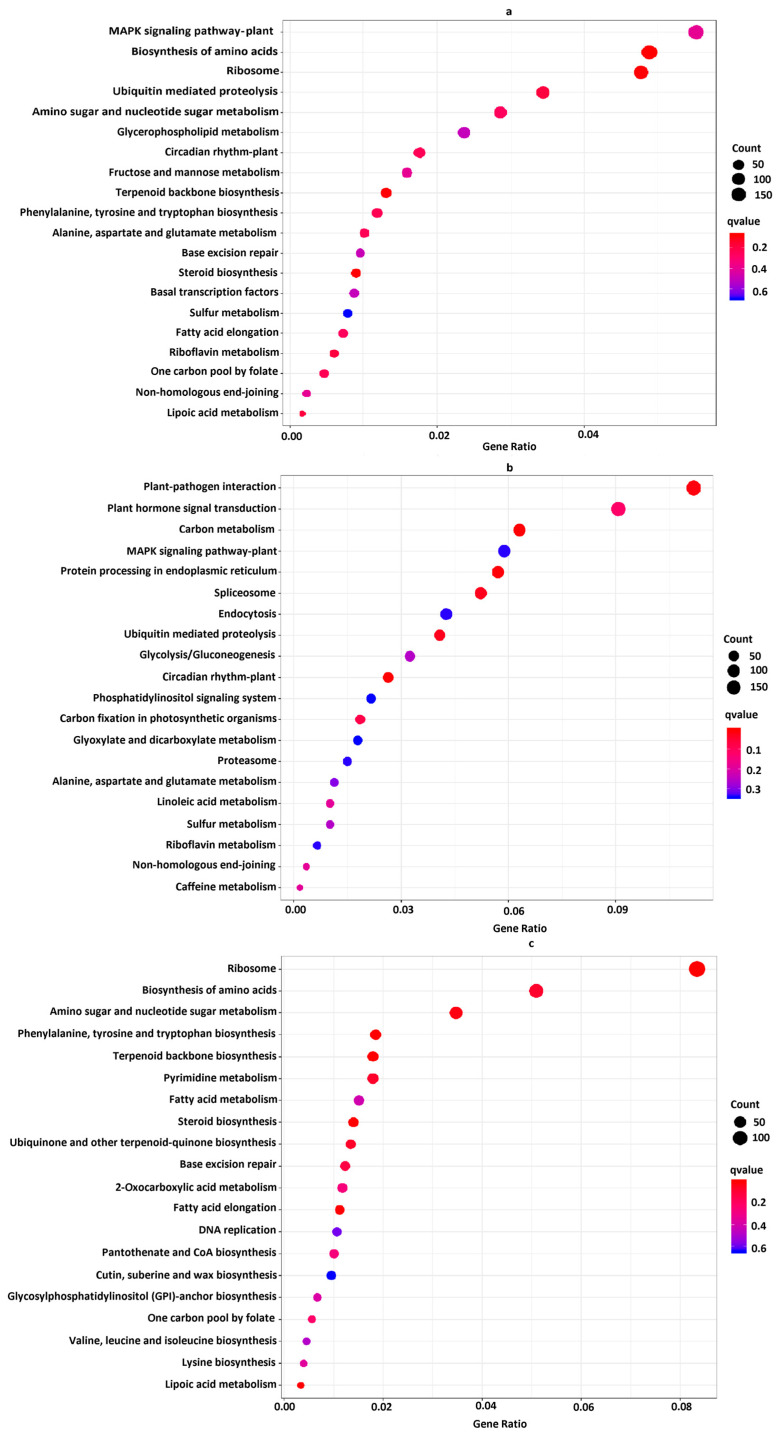
KEGG pathway enrichment analysis of differential expression genes (DEGs) in LT_30_ vs. CK_30_. (**a**) KEGG enriched bubble diagram of all DEGs. (**b**) KEGG enriched bubble diagram of the up-regulated DEGs. (**c**) KEGG enriched bubble diagram of the down-regulated DEGs. The *X*-axis is the gene ratio, which indicates the proportion of interested genes in this entry to all DEGs, and the *Y*-axis is each pathway entry. The size of the dots represents the number of DEGs annotated in the pathway, and the color of the dots represents the q value of the hypergeometric test.

**Figure 5 ijms-24-14563-f005:**
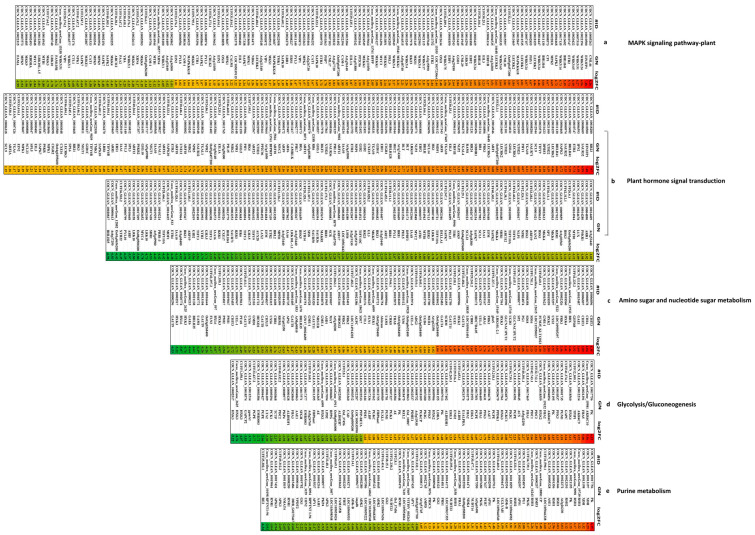
The major regulator genes of significantly enriched KEGG pathways in LT_30_ vs. CK_30_. The red color indicates up-regulated genes, the darker the color, the greater the up-regulatedgenes, and the green color indicates down-regulated genes, the darker the color, the greater the down-regulated genes (the same below) (**a**) MAPK signaling pathway−plant. (**b**) Plant hormone signal transduction. (**c**) Amino sugar and nucleotide sugar metabolism. (**d**) Glycolysis/gluconeogenesis. (**e**) Purine metabolism.

**Figure 6 ijms-24-14563-f006:**
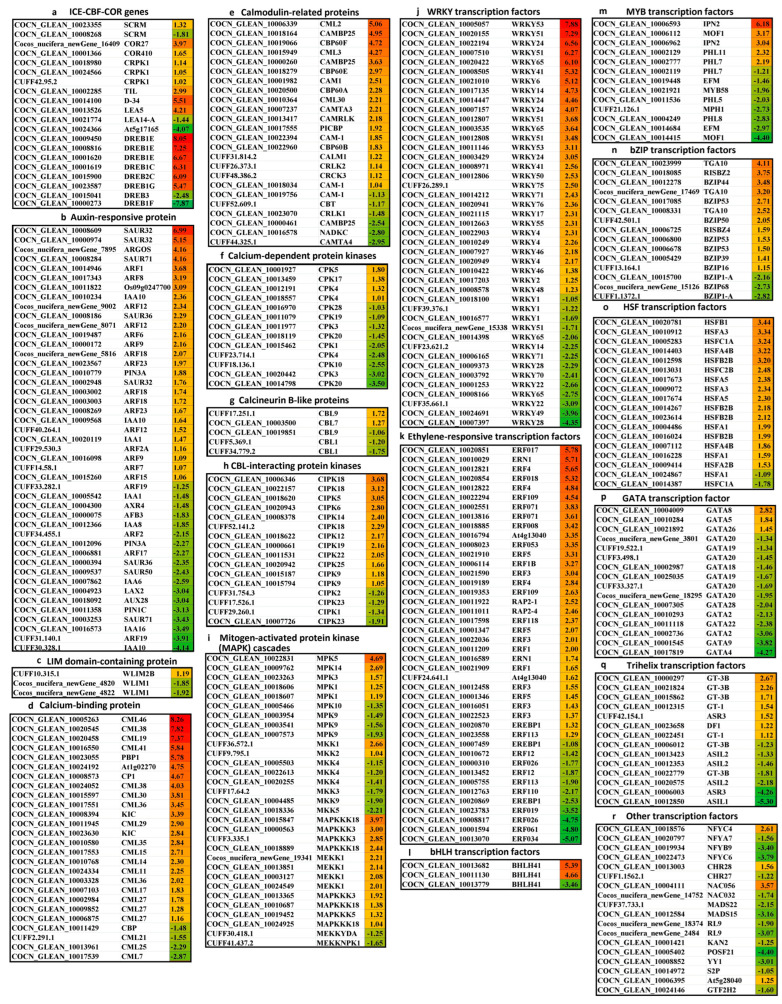
Core genes and transcription factor (TF) in LT_30_ vs. CK_30_. (**a**) Genes in ICE-CBF-COR pathway. (**b**) Genes in auxin-responsive proteins. (**c**) Genes in LIM domain-containing protein. (**d**) Genes in calcium-binding proteins. (**e**) Genes in calmodulin-related proteins. (**f**) Genes in calcium-dependent protein kinases. (**g**) Genes in calcineurin B-like proteins. (**h**) Genes in CBL-interacting protein kinases. (**i**) Genes in MAPK cascades. (**j**) Genes in WRKY TFs. (**k**) Genes in ethylene-responsive TFs. (**l**) Genes in bHLH TFs. (**m**) Genes in MYB TFs. (**n**) Genes in bZIP TFs. (**o**) Genes in HSF TFs. (**p**) Genes in GATA TFs. (**q**) Genes in trihelix TFs. (**r**) Genes in other TFs.

**Figure 7 ijms-24-14563-f007:**
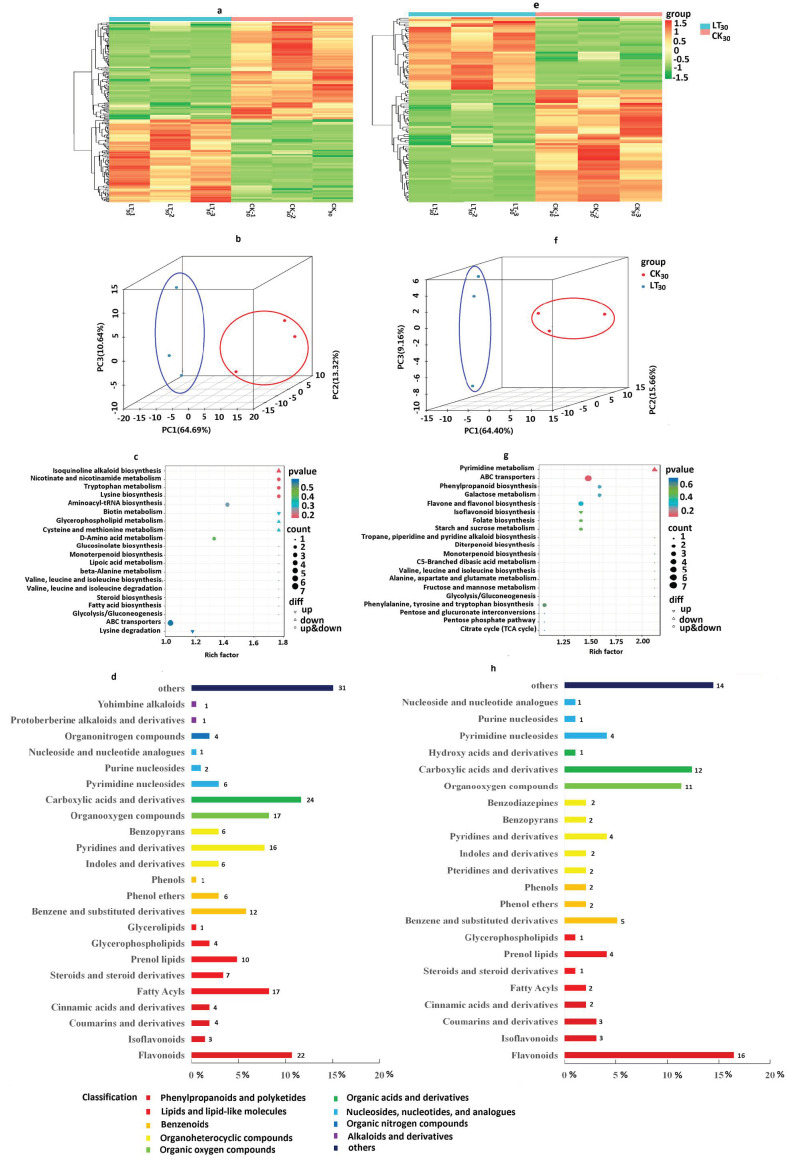
Metabolome analysis in LT_30_ vs. CK_30_. (**a**,**e**) Clustered heat maps of differentially accumulated metabolites (DAMs) in the positive- and negative-ion modes. The *X*-axis represents each sample, and the *Y*-axis represents the quantitative value of metabolites standardized by Z-score after hierarchical clustering. Color bar marks distinguish different groups. (**b**,**f**) Principal component analysis (PCA) 3D diagram of DAMs difference grouping in the positive- and negative-ion modes. The *X*-axis represents the first principal component, the *Y*-axis represents the second principal component, and the *Z*-axis represents the third principal component. The percentage axis indicates the contribution of the principal component to the sample difference. Each point represents a sample, and samples in the same group are represented with the same color, and samples in different groups are labeled with different colors. (**c**,**g**) KEGG enrichment map of top 20 DAMs in the positive- and negative-ion modes. Each point represents a KEGG pathway, and the *X*-axis is Rich_factor, which represents the ratio of differential metabolites annotated to a pathway to the ratio of all metabolites annotated to that pathway. The *Y*-axis is the path name. The shade of the dot represents *p* value. The size of the circle indicates the number of differential metabolites enriched in the pathway. The differential metabolites in the pathway are up-regulated, represented by the lower triangle, and the differential metabolites in the pathway are down-regulated, represented by the upper triangle. There are both up-regulated and down-regulated differential metabolites in the pathway, which are represented by circles. (**d**,**h**) HMDB database classification summary of metabolites in the positive- and negative-ion modes. Items in the same box in the figure represent HMDB level classification information, corresponding to the super class and class information of the HMDB database. Column length represents the number of metabolites annotated by the classification.

**Figure 8 ijms-24-14563-f008:**
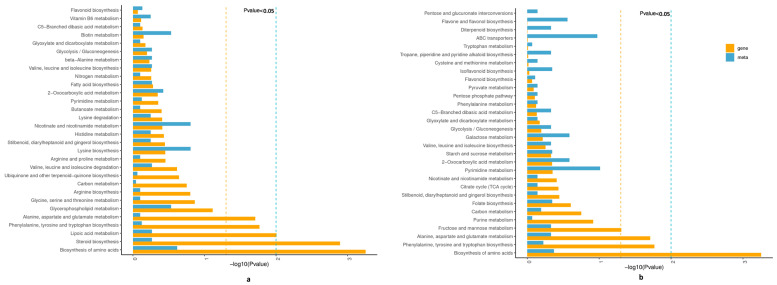
KEGG pathway enrichment analysis of top 30 differential expression genes (DEGs) and differential accumulated metabolites (DAMs) association in LT_30_ vs. CK_30_. (**a**) DEGs and DAMs significantly co-enriched KEGG pathways in the positive mode. (**b**) DEGs and DAMs significantly co-enriched KEGG pathways in the negative-ion mode. Each column represents a KEGG pathway, and different colors represent different omics, with yellow representing the transcriptome and blue representing the metabolome. The ordinate text indicates the name of the path. The horizontal coordinate represents the significance of the path enrichment, that is, the FDR value, and the logarithm of FDR is taken.

**Figure 9 ijms-24-14563-f009:**
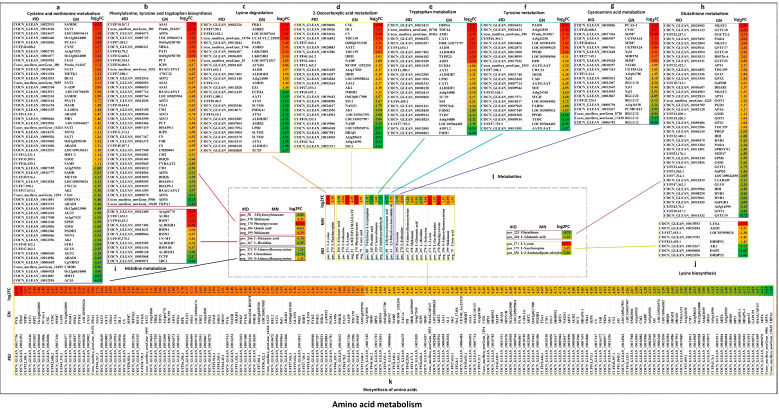
Comprehensive analysis of metabolomics and transcriptomics of coconut seedlings under cold stress in amino acid metabolism, including pathways such as (**a**) changes in DEGs in cysteine and methionine metabolism; (**b**) changes in DEGs in phenylalanine, tyrosine, and tryptophan biosynthesis; (**c**) changes in DEGs in lysine degradation; (**d**) changes in DEGs in 2-Oxocarboxylic acid metabolism; (**e**) changes in DEGs in tryptophan metabolism; (**f**) changes in DEGs in tyrosine metabolism; (**g**) changes in DEGs in cyanoamino acid metabolism; (**h**) changes in DEGs in glutathione metabolism; (**i**) changes in DEGs in histidine metabolism; (**j**) changes in DEGs in lysine biosynthesis; and (**k**) changes in DEGs in biosynthesis of amino acids. (**l**) Changes in metabolites associated with these pathways.

**Figure 10 ijms-24-14563-f010:**
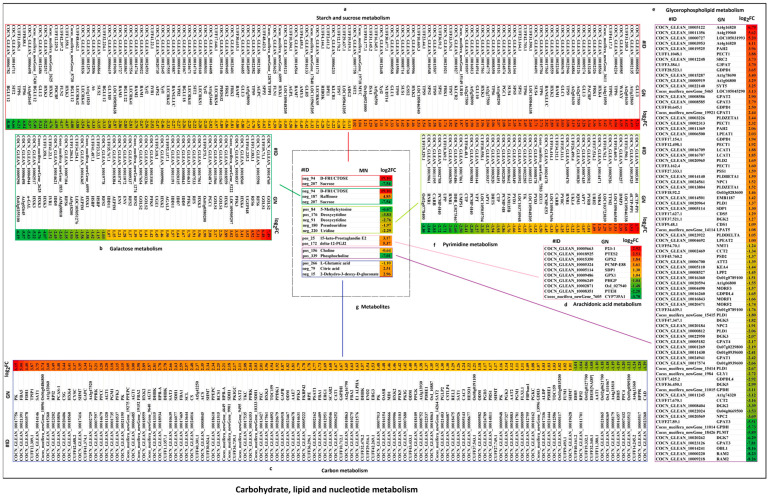
Comprehensive analysis of metabolomics and transcriptomics of coconut seedlings under cold stress in carbohydrate, lipid, and nucleotide metabolism, including the pathways such as (**a**) changes of DEGs in starch and sucrose metabolism, (**b**) changes of DEGs in galactose metabolism, (**c**) changes of DEGs in carbon metabolism, (**d**) changes of DEGs in arachidonic acid metabolism, (**e**) changes of DEGs in glycerophospholipid metabolism, (**f**) changes of DEGs in pyrimidine metabolism, and (**g**) changes in metabolites associated with these pathways.

## Data Availability

The raw RNA-seq data can be found at NCBI SRA under the project number: PRJNA974415, SRR24659939.

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
