# Peer review of "Integrated Transcriptomic and Metabolomics Analyses Reveal Molecular Responses to Cold Stress in Coconut (Cocos nucifera L.) Seedlings"

_ijms, 2023, doi:10.3390/ijms241914563_

Round 1

Reviewer 1 Report

Dear Authors,

A very comprehensive work with a rich methodological workshop. What I miss in the methods is a detailed description of the methods used to determine the activity of enzymes. Obviously , I understand that you used a ready-made kit , so please give its catalog number , so that the reader (scientist) can also apply such methods.  Similarly , with metabolomics - what standards and of what purity were used in the identification of these compounds. With such a multitude of methods , it is necessary to take care of a well-written methodology . Only then can the correctness of these methods be evaluated. 

Author Response

Review Report 1

A very comprehensive work with a rich methodological workshop. What I miss in the methods is a detailed description of the methods used to determine the activity of enzymes. Obviously , I understand that you used a ready-made kit , so please give its catalog number , so that the reader (scientist) can also apply such methods.  Similarly , with metabolomics-what standards and of what purity were used in the identification of these compounds. With such a multitude of methods , it is necessary to take care of a well-written methodology . Only then can the correctness of these methods be evaluated.

Revision:Thanks for your advice, we refer to SOD (A001-3-2), CAT(A007-1-1), and POD (A084-3-1) kit instructions and Li method by purchasing kit, extraction and detection method. Detailed extraction and detection are as follows. After extraction, enzymogram detection, seen in line in red mark. And metabolomics analysis,The raw data collected using MassLynx V4.2 was processed using Progenesis QI software for peak extraction, peak alignment, and other data processing operations, the data was identified based on Progenesis QI software, the online METLIN database and the self built database of Baimaike (http://www.biomarker.com.cn/) , and theoretical fragment identification was performed. The quality deviation was within 100ppm, seen in line in red mark.

Reviewer 2 Report

The manuscript presents a great amount of new and interesting data, but presenting in a very confuse manner, so it is very diffivult to grasp some information. It has to be rewritten and extensively shortened in order to meet the qulity standards of IJMS.

Specifically:

Introduction:

The papers deals with chilling stress, not freezing stress. Even though in some parts the authors explain the effects of freezing stress (lines 67-70). Please delete and focus on chilling stress.

Authors summarize similar studies performed in most horticultural crops, but do not mention studies made in other trees such as:

Taïbi, K., et al. Distinctive physiological and molecular responses to cold stress among cold-tolerant and cold-sensitive Pinus halepensis seed sources. BMC Plant Biol 18, 236 (2018). https://doi.org/10.1186/s12870-018-1464-5

Results: 

The results are written in a tedious way, just enumerating all the categories. These data should be visualized in the figures, and there is no point in mentioning each and every one of the categories. For instance: lines 242-265; 275-318; 328-441 or 614 to 704. Please refer only to the most represented and try to comment the most relevant information, not all the information. 

Line 715: Not all the amino acids are considered osmolites. Mainly proline and arginine. Please correct.

Figure 3: Enlarge the lettering or split into different figures.

Discussion: Again, long and tedious, and just a recitation of the data that can be found in the figures. Please rewrite it to comment the most relevant information. 

I recommend the authors to perform an additional statistical analysis (PCA or similar), like the one presented in figure 7, but  analyzing all the data in order to identify whether the data creates clusters and this will clarify the conclusion, which is the present version, is again a long list. 

I also found that there is no comment on the role of aquaporins, that are bona fide participants in the cold response at the molecular level. Have they been found in the analysis?. 

see for instance:

https://www.frontiersin.org/articles/10.3389/fpls.2018.00282/full

or

https://onlinelibrary.wiley.com/doi/10.1111/pce.13416

English is fine, no issues detected

Author Response

Review Report 2

The manuscript presents a great amount of new and interesting data, but presenting in a very confuse manner, so it is very difficult to grasp some information. It has to be rewritten and extensively shortened in order to meet the qulity standards of IJMS.

Revision:Thank you for your suggestions. I have revised the paper according to your suggestions and raised it to a language professional company for language polishing. During the revision, I have reduced some repeated sentences, rewritten some sentences, retained important information, and marked with red letters to indicate where we have revised.

Specifically:

Introduction:

  1. The papers deals with chilling stress, not freezing stress. Even though in some parts the authors explain the effects of freezing stress (lines 67-70). Please delete and focus on chilling stress.

Revision: Thanks to your suggestion, we have removed the relevant sentences on the effects of freezing stress. See lines 67-70 in revision manuscript.

  1. Authors summarize similar studies performed in most horticultural crops, but do not mention studies made in other trees such as:

Taïbi, K., et al. Distinctive physiological and molecular responses to cold stress among cold-tolerant and cold-sensitive Pinus halepensis seed sources. BMC Plant Biol 18, 236 (2018). https://doi.org/10.1186/s12870-018-1464-5

Revision:Thanks for your suggestion, we have deleted some superfluous literature and content about horticultural plants and crops, but we still retain some literature and content related to cold stress research, because some plants have common cold stress response mechanisms, including cold stress related research literature of Arabidopsis, which is of important reference value and significance for this study. Other trees cited in the introduction and discussion include Citrus reticulata,Magnolia liliiflora,Argyranthemum frutescens,apple (M. domestica), Eriobotrya japonica,oil palm, coconut and so on. The literature you recommended is about the study on physiology and molecular reaction of cold stress in Pinus halepensis, which has good reference value for this study. We also cite the literature you recommended in the introduction part, as shown in the reference number [26].

  1. Taïbi, K.;Campo, A. D. D.; Vilagrosa ; Bellés J. M.; López-Gresa, M. P.; López-Nicolás, J. M.; Mulet, J.M. Distinctive physiological and molecular responses to cold stress among cold-tolerant and cold-sensitive Pinus halepensis seed sources. BMC Plant Biol. 2018,18, 236. https://doi.org/10.1186/s12870-018-1464-5

Results: 

  1. The results are written in a tedious way, just enumerating all the categories. These data should be visualized in the figures, and there is no point in mentioning each and every one of the categories. For instance: lines 242-265; 275-318; 328-441 or 614- Please refer only to the most represented and try to comment the most relevant information, not all the information. 

Revision: Thanks for your suggestion, I have deleted lines 242-265, 275-318, 328-441 or 614-704 according to your suggestion, so as to retain important information and make readers more clear about the results of this study.seen in lines 231-259,270-304, 314-349,632-731 in

  1. Line 715: Not all the amino acids are considered osmolites. Mainly proline and arginine. Please correct.

Revision: Thanks for your suggestion, we have corrected it, see Line 715: In plants, some amino acids such as proline and arginine as osmotic substances are critical regulatory elements in stress responses.

  1. Figure 3: Enlarge the lettering or split into different figures.

Revision:Thanks for your suggestion, we have enlarged the letters as shown in Figure 3

  1. Discussion: Again, long and tedious, and just a recitation of the data that can be found in the figures. Please rewrite it to comment the most relevant information. 

Revision: Thanks for your suggestion, we have rewritten the discussion section in response to your comments, deleted some repeated sentences and relevant information, and made changes in the red font in the discussion section

  1. I recommend the authors to perform an additional statistical analysis (PCA or similar), like the one presented in figure 7, but  analyzing all the data in order to identify whether the data creates clusters and this will clarify the conclusion, which is the present version, is again a long list. 

Revision:Thank you for your suggestion. Figure 7 is about Metabolome analysis in LT30 vs. CK30. Where (a, e) Clustered heat maps of differential accumulated metabolite (DAMs) in the positive and negative ion modes.(b, f)Principal component analysis (PCA) 3D diagram of DAMs difference grouping in the positive and negative ion modes. (c, g) KEGG enrichment map of top 20 DAMs in the positive and negative ion modes. (d, h)HMDB database classification summary of metabolites in the positive and negative ion modes.  These four sets of diagrams are important for Metabolome analysis, so we confirm to retain them. In addition, I changed the original planar PCA diagram into 3D diagram, so as to better understand the principal components of the two groups of experimental data.

  1. I also found that there is no comment on the role of aquaporins,that are bona fide participants in the cold response at the molecular level. Have they been found in the analysis?. 

see for instance: https://www.frontiersin.org/articles/10.3389/fpls.2018.00282/full or https://onlinelibrary.wiley.com/doi/10.1111/pce.13416

Revision:Thanks for your suggestion, I re-searched the data and found aquaporins in the differential DEGs. Moreover, we added analysis and discussion about aquaporins in the discussion section, such as: some genes of aquaporins (TIP1-1, PIP2-4, SIP2-1, PIP2-2, NIP1-1, PIP1-2), were significantly down-regulated,  and the gene of aquaporin (SIP1-2) was significantly up-regulated, as shown in line666 in red mark. In addition, the two papers cited in the discussion are numbered [89,90].

89.He, W.D.; Gao, J.; Dou, T.X.; Shao, X.H.; Bi, F.C.; Sheng, O.; Deng, G.M.; Li, C.Y.; Hu, C.H.; Liu, J.H.; et al. Early Cold-Induced Peroxidases and Aquaporins Are Associated With High Cold Tolerance in Dajiao (Musa spp. ‘Dajiao’). Front. Plant Sci. 2018, 9, 282. http://doi.org/10.3389/fpls.2018.00282

90.Rosa, P.; Antonio, B.; Roc, Ros.; Ramón, S.; José, M. M. S. BvCOLD1: A novel aquaporin from sugar beet (Beta vulgaris L.) involved in boron homeostasis and abiotic stress.Plant Cell Environ. 2018, 41, 2844–2857.http://doi.org/10.1111/pce.13416

Round 2

Reviewer 2 Report

Authors hav made a great effort to improve the manuscript.

I will recommend publication.